



# Source apportionment of atmospheric PM$_{10}$ Oxidative Potential: synthesis of 15 year-round urban datasets in France

Samuël Weber[1], Gaëlle Uzu[1], Olivier Favez[2,3], Lucille Joanna Borlaza[1], Aude Calas[1], Dalia Salameh[1], Florie Chevrier[1,4,8], Julie Allard[1], Jean-Luc Besombes[4], Alexandre Albinet[2,3], Sabrina Pontet[5], Boualem Meshbah[6], Grégory Gille[6], Shouwen Zhang[7], Cyril Pallares[9], Eva Leoz-Garziandia[2,3], and Jean-Luc Jaffrezo[1]

[1]Univ. Grenoble Alpes, CNRS, IRD, IGE (UMR 5001), 38000 Grenoble, France
[2]INERIS, Parc Technologique Alata, BP 2, 60550 Verneuil-en-Halatte, France
[3]Laboratoire Central de Surveillance de la Qualité de l'air, 60550 Verneuil-en-Halatte, France
[4]Univ. Savoie Mont Blanc, CNRS, EDYTEM, 73000 Chambéry, France
[5]Atmo Auvergne-Rhône-Alpes, 69500 Bron, France
[6]Atmo Sud, 13294 Marseille, France
[7]Atmo Hauts de France, 59044 Lille, France
[8]Atmo Nouvelle Aquitaine, 33692 Mérignac, France
[9]Atmo Grand Est, 67300 Schiltigheim, France

**Correspondence:** Gaëlle Uzu (gaelle.uzu@ird.fr) and Samuël Weber (samuel.weber@univ-grenoble-alpes.fr)

**Abstract.** Reactive oxygen species (ROS) carried or induced by particulate matter (PM) are suspected to induce oxidative stress in vivo, leading to adverse health impacts, such as respiratory or cardiovascular diseases. The oxidative potential (OP) of PM, displaying the ability of PM to oxidize the lung environment, is gaining a strong interest to examine health risks associated to PM exposure. In this study, OP was measured by two different acellular assays (dithiothreitol, DTT and ascorbic acid, AA)

on PM$_{10}$ filter samples from 15 yearly time series of filters collected at 14 different locations in France between 2013 and 2018, including urban, traffic and Alpine valley site typologies. A detailed chemical speciation was also performed on the same samples allowing the source-apportionment of PM using positive matrix factorization (PMF) for each series, for a total number of more than 1700 samples. This study provides then a large-scale synthesis on the source-apportionment of OP using coupled PMF and multiple linear regression (MLR) models. The primary road traffic, biomass burning, dust, MSA-rich, and

primary biogenic sources had distinct positive redox-activity towards the OP$^{DTT}$ assay, whereas biomass burning and road traffic sources only display significant activity for the OP$^{AA}$ assay. The daily median source contribution to the total OP$^{DTT}$ highlighted the dominant influence of the primary road traffic source. Both the biomass burning and the road traffic sources contributed evenly to the observed OP$^{AA}$. Therefore, it appears clearly that residential wood burning and road traffic are the two main target sources to prioritized in order to decrease significantly the OP in Western Europe and, would the OP being a

good proxy of human health impact, to lower the health risks from PM exposure.



# 1 Introduction

Air quality has become a major public health issue, being considered as the fourth global cause of mortality with 7 million premature deaths worldwide per year due to both indoor and outdoor exposure (World Health Organization, 2016). Possibly
driving 90% of this health impact (Lelieveld et al., 2015), particulate matter (PM) is one of the key pollutants in the air linked to health outcomes, although the exact mechanism leading to toxicity is not yet fully understood (Barraza-Villarreal et al., 2008; Beck-Speier et al., 2012; Brauer et al., 2012; Goix et al., 2014; Goldberg, 2011; Saleh et al., 2019). Many urbanized areas, mainly located in low- or middle-incomes countries, are exposed to particulate matter (PM) concentration far higher than the recommendation guideline of the WHO.

Although PM are now monitored in most countries and large efforts are observed to document ambient concentrations, the underlying processes leading to the observed concentrations in the atmosphere, and particularly the understanding of emissions sources, are still active fields of research (Diémoz et al., 2019; El Haddad et al., 2011; Golly et al., 2019; Hodshire et al., 2019; Jaffrezo et al., 2005; Jiang et al., 2019; Marconi et al., 2014; Moreno et al., 2010; Piot et al., 2012; Salameh et al., 2015; Samaké et al., 2019a; Waked et al., 2014). In recent years, strong focus has been put worldwide on source-apportionment methods in
order to better understand the processes leading to the airborne concentrations and the accumulation of PM in the atmosphere. This includes direct modeling approaches such as Chemistry Transport Model (CTM) using tagged species (Brandt et al., 2013; Kranenburg et al., 2013; Mircea et al., 2020; Wagstrom et al., 2008; Wang et al., 2009) or field studies coupled with receptor models (RM) (Belis et al., 2020; Pernigotti et al., 2016; Simon et al., 2010), notably Positive Matrix Factorization (PMF). PMF can be based either on AMS time resolve spectrum (Bozzetti et al., 2017; Petit et al., 2014; Zhang et al., 2019) or on filter
analysis (Amato et al., 2016; Bressi et al., 2014; Fang et al., 2015; Jain et al., 2018, 2020; Liu et al., 2016; Petit et al., 2019; Salameh et al., 2018; Srivastava et al., 2018; Waked et al., 2014) or a mix of these different measurement techniques (Costabile et al., 2017; Vlachou et al., 2018, 2019). Results of these studies indicate that PM originates from a wide variety of sources, not only from natural (volcano, sea spray, soil dust, vegetation, bacteria, pollen. . . ) or anthropogenic (road traffic, residential heating, industry. . . ) sources, but is also formed as secondary product and condensed from the gaseous phase (ammonium-
nitrate and -sulfate. . . ). As a result, the chemistry, size distribution or reactivity of PM widely vary from location to location and season to season, which induces large changes in the health impacts depending on all of these parameters (Kelly and Fussell, 2012).

Furthermore, the mass of PM is not the most relevant metric when dealing with health impacts of airborne particles since major properties (chemistry, shape, size distribution, solubility, speciation) driving PM toxicity are not taken into account
within this single mass metric. It is now believed that the measurement of the reactive oxygen species (ROS) issued from PM, may be more closely linked to the potential adverse health effects of atmospheric PM, since oxidative stress is a key factor in the inflammatory response of the organism, leading for instance to respiratory diseases or when exposed for a long period of time, cardiovascular diseases or even cancer (Lelieveld et al., 2015; Li et al., 2003). Therefore, the oxidizing potential (OP) of PM being an indirect measure of the ability of the particles to induce ROS in a biological medium (Ayres et al., 2008; Cho
et al., 2005; Li et al., 2009; Sauvain et al., 2008) has been proposed as a potential proxy of the health impacts of atmospheric





PM exposure. Indeed, even if the clear demonstration of the OP to be a good proxy of health impact is still needed, some recent studies already established associations between OP and different possible health outcomes (Costabile et al., 2019; Karavalakis et al., 2017; Steenhof et al., 2011; Strak et al., 2017b; Tuet et al., 2017a; Weichenthal et al., 2016) or cellular stress in vitro (Leni et al., 2020).

However, there is also no consensus towards a standardized method to measure the OP of PM, and many assays and protocols co-exist (DTT, GSH, AA, ESR, °OH or $H_2O_2$, among others), with samples extracted with different methods (water, methanol, simulated lung fluid (SLF), etc.) and not always with a constant mass of PM. The dithiothreitol (DTT) and ascorbic-acid (AA) assays are widely used in associations with health endpoints (Abrams et al., 2017; Atkinson et al., 2016; Bates et al., 2015; Canova et al., 2014; Fang et al., 2016; Janssen et al., 2015; Strak et al., 2017a; Weichenthal et al., 2016; Yang et al., 2016;

Zhang et al., 2016) even if the exact methodologies differ from one study to the other. Results can also differ for the seasonality of OP based on these two assays and some studies report strong seasonality of OP whereas others don't (Bates et al., 2015; Calas et al., 2019; Cesari et al., 2019; Fang et al., 2016; Ma et al., 2018; Paraskevopoulou et al., 2019; Perrone et al., 2016; Pietrogrande et al., 2018; Verma et al., 2014; Fang et al., 2015; Weber et al., 2018; Borlaza et al., 2018; Zhou et al., 2019). Finally, several studies have already shown that different sources of PM have different reactivity to OP tests (Verma et al.,

2014; Bates et al., 2015; Fang et al., 2016; Weber et al., 2018; Paraskevopoulou et al., 2019; Cesari et al., 2019; Zhou et al., 2019; Daellenbach et al., 2020). In particular, sources with high concentrations of transition metals, such as road traffic, appear to have a higher intrinsic oxidative potential (i.e. OP per microgram of PM) than other sources of PM. However, the number of these studies is still limited and they do not always take into account complete seasonal cycles and therefore may not encompass the variety of sources for a given site, possibly omitting some important sources. Also, spatial variability at a country-scale is

currently unknown and requires homogeneous sampling and analysis methodologies for all filters and time-series.

For a comprehensive investigation of the intrinsic OP of various PM sources, we build-up an extensive dataset of about 1 700 samples from 14 sites consisting of 15 year-round time-series of observations over continental France, collected during research programs conducted between 2013 and 2018. On each of these samples, we concurrently measured the OP with the DTT and AA assays, together with an extensive chemical characterization allowing PM source apportionment using a harmonized PMF

(Positive Matrix Factorization) approach (Weber et al., 2019). Then, we apportioned the OP measured by the DTT and AA assays to the emission sources using a multilinear regression approach, following Weber et al. (2018). In this way, we can estimate the oxidizing capacity of each microgram of PM from the different identified emission sources but also the relative contribution of the different sources to the $OP^{DTT}$ and $OP^{AA}$ on seasonal and daily bases. These results are presented in this paper.

## 80    2   Materials and methods

### 2.1   Sites description

The selected sites had to fulfill three conditions: 1) a yearly sampling period, 2) the required chemical analysis to perform a harmonized PMF analysis and 3) enough filter surface left to assess the OP measurements. A total of 14 sites were included





in this study (one being sampled twice at 5 years interval) taken from different research programs. These sites reflect the

diversity of typology we could encounter in the Western Europe: urban (NGT, TAL, AIX, MRS-5av, NIC), urban alpine valley (GRE-cb, GRE-fr, VIF, CHAM, MNZ, PAS), industrial (PdB) and traffic (RBX & STG-cle) (see supplementary information (SI) Table S1) covering different areas of France (Figure 1). We can note, however, the absence of remote or rural sites in our current dataset. The Air Quality at all of these sites is monitored by the local air quality agencies (Atmo Sud, Atmo Auvergne Rhône-Alpes, Atmo Nouvelle Aquitaine, and Atmo Hauts de France) (Favez et al., 2021).

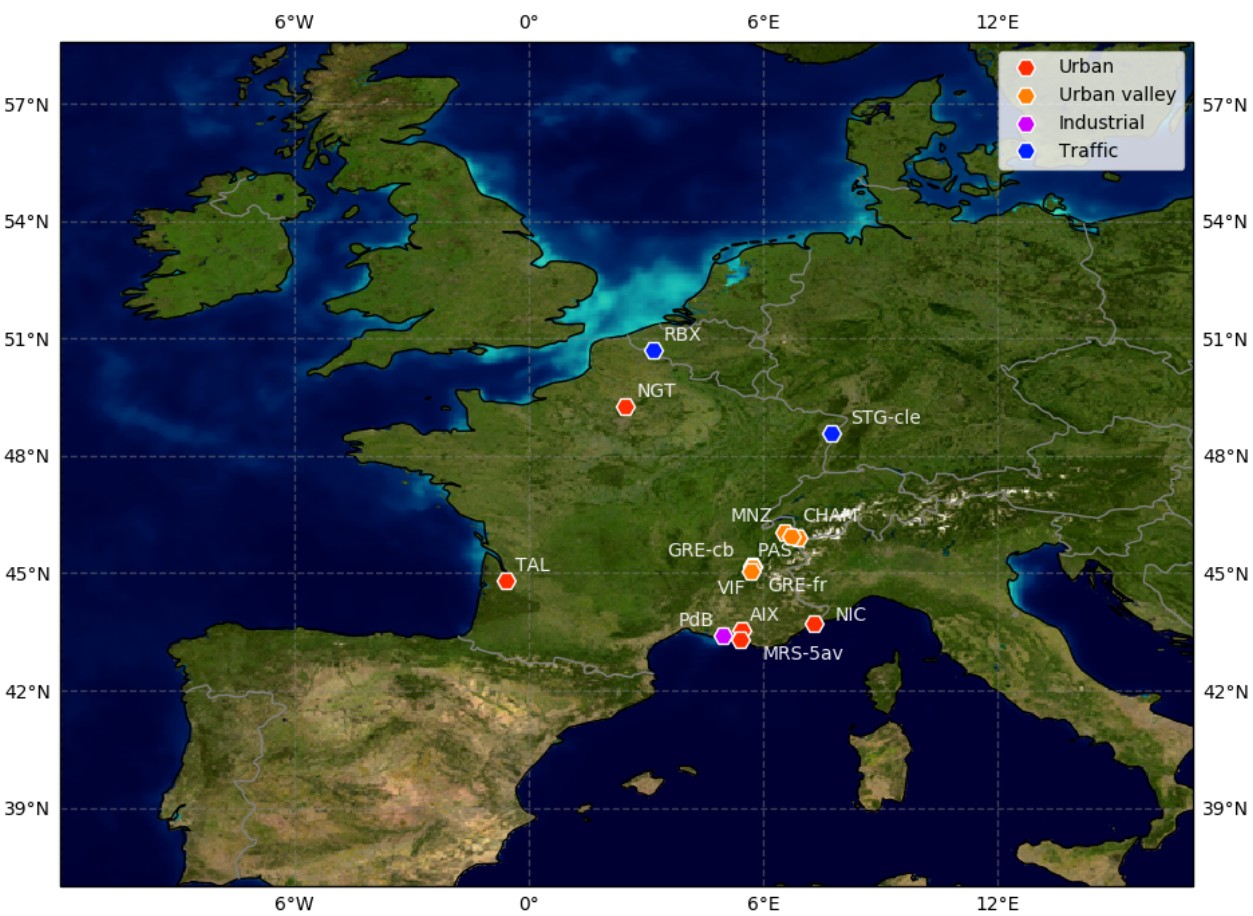

**Figure 1.** Location of the 14 sampling sites. Color codes denote the typology of the site: red, urban; orange, urban valley; magenta, industrial; blue, traffic. Background photography from NASA's Earth Observatory.





## 2.2 Sample analysis

### 2.2.1 Chemical speciation

The $PM_{10}$ concentrations were measured at each site by means of an automatic analyzer, according to EN 16450:2017 (CEN 2017b), and daily (24 hours) filter samples were collected every third day. Samplings were achieved on pre-heated quartz fiber filters using high-volume sampler (DA80, Digitel), following EN 12341:2014 procedures (CEN 2014). Off-line chemical analysis performed on these filters have been fully described previously (Weber et al., 2019). Briefly, the elemental and organic carbon fractions (EC and OC) were measured via thermo-optical analysis (Sunset Lab. Analyzer (Birch and Cary, 1996)) using the EUSAAR-2 protocol (Cavalli et al., 2010; CEN, 2017a). Major water-soluble inorganic contents ($Cl^-$, $NO_3^-$, $SO_4^{2-}$, $NH_4^+$, $Na^+$, $K^+$, $Mg^{2+}$, and $Ca^{2+}$) and methanesulfonic acid (MSA) were determined using ion chromatography (CEN, 2017b; Jaffrezo et al., 2005). Metals or trace elements (e.g., Al, Ca, Fe, K, As, Ba, Cd, Co, Cu, La, Mn, Mo, Ni, Pb, Rb, Sb, Sr, V, and Zn) were measured by inductively coupled plasma atomic emission spectroscopy or mass spectrometry (ICP-AES or ICP-MS) (Alleman et al., 2010; Mbengue et al., 2014; CEN, 2005). Finally, anhydrosugars and sugar alcohols (including levoglucosan, mannosan, arabitol, sorbitol, and mannitol) were analyzed using liquid chromatography followed by pulsed amperometric detection (LC-PAD) (Verlhac et al., 2013; Yttri et al., 2015).

### 2.2.2 OP assays

Identical methodologies were applied for all the OP measurements of the collected filters (Calas et al., 2017, 2018, 2019). The extraction of PM was performed using a simulated lung fluid (SLF: Gamble + DPPC) to simulate the bio-accessibility of PM and to closely simulate exposure conditions. In order to take into account the non-linearity of the OP with PM mass and to have comparable results between sites, the extraction has been carried out at iso-mass concentration (10 $\mu g\,ml^{-1}$ or 25 $\mu g\,ml^{-1}$ of PM, depending on the site, both values being in the low mass range of linear dose-response), by adjusting the surface of filter extracted. The filter extraction method allowed to include both soluble and insoluble particles into the extracts. After the SLF extraction, particles removed from filter were not filtrated, the whole extract was injected in a 96-wells plate for analysis. Samples were processed using the AA and DTT assays, as described below.

DTT depletion when in contact with PM extracts was determined by dosing the remaining amount of DTT with DTNB (dithionitrobenzoic acid) at different reaction times (0, 15 and 30 minutes) and absorbency was measured at 412 nm using a plate spectrophotometer (Tecan, M200 Infinite). The AA assay is a simplified version of the synthetic respiratory tract lining fluid (RTFL) assay (Kelly and Mudway, 2003), where only AA is used. AA depletion was read continuously for 30 minutes by absorbency at 265 nm (TECAN, M1000 Infinite). The depletion rate of AA was determined by linear regression of the linear section data. For both assays, the 96-wells plate was auto shaken for 3 seconds before each measurement and kept at 37 °C. Three filter blanks (laboratory blanks) and three positive controls (1,4-napthoquinone, 24.7 $\mu mol\,l^{-1}$) were included in each plate ($OP^{AA}$ and $OP^{DTT}$) of the protocol. The average values of these blanks were then subtracted from the sample measurements of the given plate. Detection limit (DL) value were defined as three times the standard deviation of laboratory blank measurements (laboratory blank filters in Gamble+DPPC solution).





Hereafter, the OP$^{DTT}$ and OP$^{AA}$ normalized by air volume are noted OP$_v^{DTT}$ and OP$_v^{AA}$, respectively, with unit of $\mathrm{nmol\,min^{-1}\,m^{-3}}$.

## 2.3 Source apportionment

The source apportionment of the OP can be performed in two main ways: 1) by including the OP as an input variable for receptor-model (RM) (Verma et al., 2014; Fang et al., 2016; Ma et al., 2018; Cesari et al., 2019) or 2) by conducting source attribution to the PM mass and then, using a multiple linear regression (MLR) model, assigning OP to each of the sources from the source-receptor model (Bates et al., 2015; Verma et al., 2015b; Weber et al., 2018; Cesari et al., 2019; Paraskevopoulou et al., 2019; Zhou et al., 2019; Daellenbach et al., 2020). We decided to use the second approach since adding the OP variable

to the PMF may change the source apportionment solution. Further, the first method would imply that the intrinsic OP would be positive by construction and not due to relevant physical properties (see below). Moreover, the OP apportionment in a 2-steps process (PM mass source apportionment then OP apportionment) allowed to potentially use different model types for the OP apportionment and re-use results of the PMF for others OP assays later on.

### 2.3.1 PM mass apportionment: Positive Matrix Factorization

### 2.3.2 Methodological background

The PM mass source apportionment for the 15 yearly series was conducted using the U.S. Environnmental Protection Agency (US-EPA) EPA PMF 5.0 software (US EPA, 2017) with the ME-2 solver from (Paatero, 1999). Briefly, the PMF was introduced by Paatero and Tapper (1994) and is now one of the most common approaches used for PM source-apportionment studies (Hopke et al., 2020; Karagulian et al., 2015; Belis et al., 2020). It aims at solving the receptor model equation Eq. 1

$$X = G \cdot F, \tag{1}$$

where $X$ is the $n \times m$ observation matrix, $G$ is the $n \times p$ contribution matrix and $F$ is the $p \times m$ factor profile matrix (or *source*, despite some factors not being proper emission *sources* but may reflect secondary processes), with $n$ the number of samples, $m$ the number of measured chemical species and $p$ the number of profiles. Hereafter, the $G$ matrix is expressed in $\mathrm{\mu g\,m^{-3}}$ and the $F$ matrix in $\mathrm{\mu g\,\mu g^{-1}}$ of PM.

### 2.3.3 PMF set up

Some of the PMF analyses included in this paper have been performed during previous programs, namely SOURCES (http://pmsources.u-ga.fr, Weber et al. (2019)), DECOMBIO (Chevrier, 2016; Chevrier et al., 2016), or MobilAir (https://mobilair.univ-grenoble-alpes.fr/, Borlaza et al. (2020)). In order to get comparable PM sources profiles from a common set of input species and constraints in the model, all PMF analyses have been ran again for this study according to a harmonized method-

ology, as previously reported (Weber et al., 2019).

The input species were slightly different from one study to another, but always include carbonaceous compound (OC and EC), ions (SO$_4^{2-}$, NO$_3^-$, Cl$^-$, NH$_4^+$, K$^+$, Mg$^{2+}$, Ca$^{2+}$), organic compounds (levoglucosan, mannosan, arabitol and manitol (the





latter two summed and referred to polyols) and MSA), and a set of trace metals for a total of about 30 species. The list of metals
used within the PMF analysis was not the same for each of the sites, due to too low concentrations (lower than quantification
limit) on some filters leading to a signal over noise ratio very low (see Table S2). The uncertainties were estimated following
the method proposed by Gianini et al. (2012) and were tripled if the signal over noise ratio was below 2 (classified as "weak"
in the PMF software). Between 8 to 10 factors were identified at the different sites and are summarized in Table S3. For each of
the PMF analysis, the possibility of using constraints to refine some of the chemical profile of factors was considered, in order
to better disentangle possible mixing between factors and reduce the rotational ambiguity, based on *a priori* expert knowledge
of the geochemistry of the sources. A PMF solution was considered valid if it followed the recommendation of the "European
guide on air pollution source apportionment with receptor models" (Belis et al., 2019), also requiring a proper geochemical
identification of the various factors. Estimation of the uncertainties of the PMF was obtained on both the base and constrained
runs using the bootstrap (BS) and displacement (DISP) functions of the EPA PMF5.0 (Brown et al., 2015).

### 2.3.4 Similarity assessment of the PMF factors

Since PMF resolved sites-specific PM factors, we checked if a given factor had consistent chemical profile over the different
sites. For this purpose, as presented in Weber et al. (2019), a similarity assessment of all PMF factor profiles was run following
the DeltaTool approach (Pernigotti and Belis, 2018). Using this tool, we compared pairs of factor profiles based on their mass-
normalized chemical compounds using 2 different metrics, namely the Pearson distance (PD) and the standardized identity
distance (SID) (Belis et al., 2015). The first one defined as Eq. 2

$$PD = 1 - r^2 \tag{2}$$

with $r^2$ the Pearson correlation coefficient, might be strongly influenced by individual extreme points. The second one, SID,
expressed as follows (Eq. 3):

$$SID = \frac{\sqrt{2}}{m} \sum_{j=1}^{m} \frac{|x_j - y_j|}{x_j + y_j}, \tag{3}$$

where $x$ and $y$ are two different factors profiles expressed in relative mass, and $m$ the number of common specie in $x$ and $y$, is
evenly sensitive to every species since it includes a normalization term.

### 2.4 OP apportionment

The computation was done thanks to the *statsmodels* 0.12 python package (Seabold and Perktold, 2010) and the graphics were
produced with *matplotlib* 3.3.1 (Hunter, 2007; Caswell et al., 2020) and *seaborn 0.11* (Waskom and the seaborn development
team, 2020).

### 2.4.1 Apportionnement using Multi linear regression (MLR)

MLR was conducted independently at each site, with results from the two (DTT and AA) OP assays being the dependent
variables and the sources contribution obtained from the PMF being the explanatory variables, following the equation Eq. 4,



similar to Weber et al. (2018):

$$OP_{obs} = G \times \beta + \varepsilon,\tag{4}$$

where $OP_{obs}$ is a vector of size $n \times 1$ of the observed $OP_v^{DTT}$ or $OP_v^{AA}$ in $\mathrm{nmol\,min^{-1}\,m^{-3}}$, $G$ is the matrix $(n \times (p+1))$ of the mass contribution of PM sources obtained from the PMF in $\mathrm{\mu g\,m^{-3}}$ and a constant unit term for the intercept (no unit), $\beta$ are the coefficients (i.e. intrinsic OP of the source and the intercept) of size $((p+1) \times 1)$ in $\mathrm{nmol\,min^{-1}\,\mu g^{-1}}$ for the intrinsic OP and in $\mathrm{nmol\,min^{-1}\,m^{-3}}$ for the intercept. The residual term $\varepsilon$ $(n \times 1)$ accounts for the misfit between the observations and the model.

The intercept was not forced to zero on purpose. Indeed, if the system is well constrained the intercept should spontaneously be close to zero and conversely a non-zero intercept would point out missing explanatory variables.

A weighted least square regression (WLS) was finally used to consider the uncertainties of the OP measurements. The uncertainties of the coefficients $\beta$ given by the MLR were estimated by bootstrapping the solutions 500 times, randomly selecting 70% of the samples each time to account for possible remaining extremes events or seasonal variations of the intrinsic

OP per source. The uncertainty of the PMF result G was however not considered because the EPA PMF software only returns to the user the uncertainties associated with the profile matrix F (see (Weber et al., 2019) for a first order estimation of the G uncertainties).

### 2.4.2 Contribution of the sources to the OP

The contribution $G^{OP}$ in $\mathrm{nmol\,min^{-1}\,m^{-3}}$ of the sources to the OP was computed at each site independently and was calculated

following Eq. 5

$$G_k^{OP} = G_k \times \beta_k\tag{5}$$

where $k$ is the source considered, G the PMF sources' contribution in mass concentration ($\mu g/m^3$) and $\beta$ the intrinsic OP of the sources in $\mathrm{nmol\,min^{-1}\,\mu g^{-1}}$. The uncertainties of $G^{OP}$ were computed using to the uncertainties of $\beta$ estimated from the 500 bootstraps.

### 2.5  Focus on the main PMF factors

This study focuses on the main drivers of OP at the regional scale. For this reason, we decided to include in the main discussion only the PMF factors identified at least in two-thirds of the series (i.e. 10 out of 15 series), namely the aged salt, biomass burning, dust, MSA-rich, nitrate-rich, primary biogenic, primary road traffic and sulfate-rich factors. However, the remaining sources, often local, barely contributed to the total PM mass and important uncertainties were often attached to them. The only

notable exception is the HFO (heavy fuel oil) profile identified at some coastal sites, discussed hereafter in its own section.





## 3   Results and discussion

As a large set of results has been obtained in the present study and cannot be exhaustively presented here, an interactive visualization tool providing details on PM and OP sources time series and apportionment outputs is available online at http://getopstandop.u-ga.fr/ and is proposed as supplementary material of this manuscript including all factors for all series.

Note that, since there are more samples where PMF has been run compared to available OP measurements (around 1700 concomitant OP and PMF samples compare to 2048 samples with a PMF solution), the discussion hereafter on the sources contributions to the OP takes into account the whole PMF analysis, including days when models were not trained but predicted by the above-mentioned approach.

### 3.1   PMF results

In this section, we summarize the main results acquired from the harmonized PMF approach conducted for the present paper, but we invite the reader to refer to the previous study (Weber et al., 2019) and to the website to have a more complete view of the results.

#### 3.1.1   PMF source apportionment

The list of the identified factors at each site is given in SI Table S3 and individual profiles and time series together with un-
certainties can be found at http://getopstandop.u-ga.fr/results?component=pmf_profile_and_contribution. Table 1 summarizes the main PMF factors found at least at 10 out of the 15 series. Shortly, we obtained PMF factors corresponding to biomass burning (mainly from residential heating), primary road traffic, mineral dust, secondary inorganic nitrate-rich and sulfate-rich, salt (fresh and aged) as well as primary biogenic and MSA-rich. Some other local sources were also identified at some sites, targeting some local heavy loaded metals sources with a very low contribution to the total PM mass —supposedly linked to
industrial process— which contained a wide variety of chemicals but shared a common set of metal (Al, As, Cd, Mn, Mo, Pb, Rb, Zn). Finally, a factor related to shipping emission (namely heavy fuel oil, HFO) was obtained at some coastal sites.

**Table 1.** Main PMF factors identified (at least at two third of the series, i.e. 10 out of 15) and species used as proxy for the determination.

| Factor name | Number of sites identified | Main species used as tracers | General remarks |
| --- | --- | --- | --- |
| Biomass burning | 15 | Levoglucosan, OC, EC, $K^+$, Rb | High in alpine valley, strong seasonality |
| Nitrate rich | 15 | $NO_3^-$, $NH_4^+$ | Mostly in spring |
| Primary biogenic | 15 | Polyols (arabitol, mannitol), OC | Strong seasonality |
| Road traffic | 14 | OC, EC, Cu, Fe, Sb, Sn | Mixed exhaust and non-exhaust emission |
| MSA-rich | 14 | MSA | Strong seasonality |
| Dust | 13 | Al, Ti, $Ca^{2+}$ | Episodic, some OC is present |
| Sulfate rich | 13 | $SO_4^{2-}$, $NH_4^+$, Se | Some OC is present |
| Aged salt | 12 | $Na^+$, $Mg^{2+}$, $SO_4^{2-}$, $NO_3^-$ | Some OC is present |





### 3.1.2 PMF similarities between sites

The similarity between chemical profile composition estimated by the PD and SID metrics are presented in Figure S1. We observed a strong similarity for the main sources of PM, namely biomass burning, nitrate-rich, primary biogenic, sulfate-rich and to a lower extend road traffic. The dust, aged salt and MSA-rich were often identified and presented acceptable SID, but also showed large values for the PD metric. As the PD is sensitive to "extreme points", this translates in our case into different contributions for the chemical specie contributing most to the PM mass (mainly OC and EC). The MSA-rich is the most variable factor and a detailed analysis of its chemistry profile indicated many differences from site to site for the concentrations of EC but also $NO_3^-$ and $NH_4^+$. This factor being essentially a secondary organic factor, this variability may be explained by different formation or evolution pathways, or different level of aging. We can also point out that the industrial source had a very diverse chemical composition since it is related to different local industrial processes.

Nevertheless, the geochemical stability of the majority of PMF factors on a regional scale is good and allows to consider that these emission sources are homogeneous over France.

### 3.2 OP results and seasonality

The 15 time-series for both $OP^{DTT}$ and $OP^{AA}$ at each site are presented on the website (http://getopstandop.u-ga.fr/results?component=rd_ts). A monthly aggregated view is given in Figure 2, for $OP_v^{DTT}$ and $OP_v^{AA}$, respectively. As the dataset covered complete full years, including the influence of different PM sources with different seasonal activities, the results obtained are representative of spatio-temporal patterns of the OP's at least over France, and probably over large parts of Western Europe.

As reported previously by Calas et al. (2018, 2019), we observed a seasonality of both $OP_v^{DTT}$ and $OP_v^{AA}$, with higher OP values during the colder months (October-March) compared to the warmer months (April-September). We also noted that during the winter period, the statistical distribution of OP values did not follow a normal distribution and a significant variability was observed. This was especially the case for the sites located in the alpine area (GRE-fr, GRE-cb, VIF, CHAM, MNZ, PAS) showing stronger seasonality compared to the other locations. Such specificity was already reported previously by (Calas et al., 2019), together with some rapid variation of the $OP^{DTT}$ and $OP^{AA}$, with drastic increase or decrease within the frame of few days, similarly to the $PM_{10}$ mass concentration. This behavior may be related to the formation of thermal inversion layers in such valleys, leading to the accumulation of pollutants and the promotion of the secondary processes inducing increased formation of secondary organic aerosol (SOA) and of key organic specie, like polyaromatic quinones (Albinet et al., 2008; Tomaz et al., 2017; Srivastava et al., 2018), or HULIS (Baduel et al., 2010) having a significant impact on the OP.

Several sites exhibit much lower seasonality in the OP values, especially traffic sites (RBX and STG-cle), the urban traffic site (NIC), or the industrial one (PdB). The lack of seasonality for some sites exclude the hypothesis of the OP being driven only by synoptic meteorological parameters such as sunshine or temperature, as it would impact all the sites similarly. It is clearly the difference in PM chemical compounds and reactivity, together with the timing of emission, that induces the seasonality of OP values when it is observed.





For comparison with previous studies, the spearman correlation between chemical species, sources contributions and OP's

is also report in Annex A of this paper.

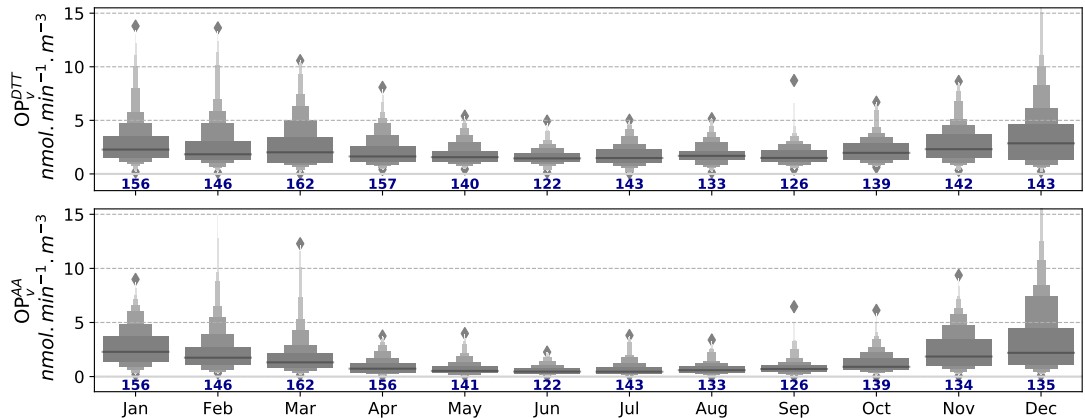

**Figure 2.** Boxenplot of $OP_v^{DTT}$ and $OP_v^{AA}$ seasonal values. The numbers in the x-axis indicate the number of observations. Each box represents one decile and the black horizontal lines indicate the median of the distributions. Some values greater than $15.5 \text{ nmol min}^{-1} \text{ m}^{-3}$ are not displayed for graphical purpose.

## 3.3 Results of OP's inversion for the main PMF sources

### 3.3.1 Model accuracy and linear limitation

The MLR statistical validation was carried out by a residual analysis between the OP observed and the OP reconstructed by the model. For this evaluation, the intrinsic OP of the sources was set to the mean of the 500 bootstrap values. All but two sites

present a very good correlation between observed and reconstructed OP ($r^2>0.7$) and a regression line close to unity (see SI Table S4, details and individual scatter plots are given at http://getopstandop.u-ga.fr/results?component=op_obsvsmodel). We therefore consider our models valid and each intrinsic OP (i.e. coefficient of the regression) may be explored individually to geochemically explain the observed OP.

However, despite our models being able to reproduce most of the observations with normally distributed residuals, it also

tends to underestimate the highest values and the residuals are often heteroscedastics (i.e. the higher values, the higher the uncertainties). Then, the underlying hypothesis of linearity between endogenous variables (PM concentration of the sources) and exogenous variables (OP's) may be deemed invalid. It is also important to note that non-linear processes are strongly suspected for the source-apportionment of OP, as already noted by Charrier et al. (2016) or Calas et al. (2018) and Samaké et al. (2017). As a result, future development on OP apportionment models should focus on this suspected non-linearity, either

by introducing co-variations terms or using non-linear models such as neural network for instance (Borlaza et al.).




### 3.3.2 Intrinsic OP of the different sources

Even if the models reproduce the observations correctly, this does not guarantee that the geochemical meaning extracted is the same for each of the models, i.e. the intrinsic OP's of the sources may completely differ from site to site. The question is then to identify if a given source contributes similarly to the OP at all sites. In other words, do all model extract any general

geochemical information relative to the OP?

Figure 3 presents the intrinsic $OP^{DTT}$ and $OP^{AA}$ for the selected subset of sources. The values of mean and standard deviation and details per station for all sources are given in Table S5, S6 and S7.

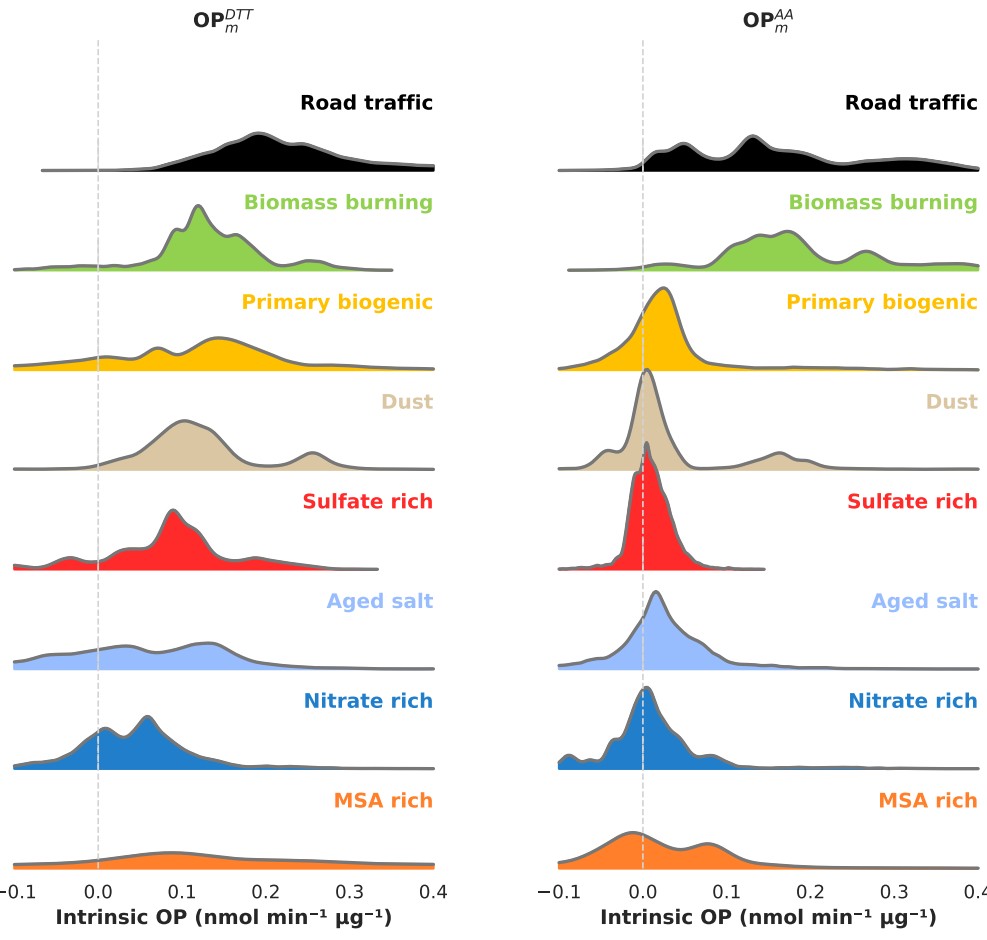

**Figure 3.** Intrinsic $OP^{DTT}$ and $OP^{AA}$ values according to the sources identified for at least two third of the site (i.e. 10 series). The number of data considered is $n \times N$ with $n$ the number of series where the source was identified and $N = 500$ bootstraps: Road traffic: 7000, Biomass burning: 7500, Dust: 6500, Primary biogenic: 7500, Nitrate-rich: 7500, Sulfate-rich: 7500, MSA-rich: 7000, Aged salt: 6000. The exact values of means and standard deviations for each site are given in the SI.





For most of the sources identified, positive average intrinsic OP values were observed considering the whole dataset. Negative mean values where only observed for the MSA-rich factor for the AA assay ($-0.018 \pm 0.152$ nmol min$^{-1}$ µg$^{-1}$) associated
with large variance. Such results highlighted again that airborne particles, whatever their sources of origin, have significant oxidative properties.

### 3.3.3 Different intrinsic OP per sources

We also observed a clear distinction between the intrinsic OP values for the different PM sources, ranging from $0.044 \pm 0.064$ nmol min$^{-1}$ µg$^{-1}$ to $0.223 \pm 0.085$ nmol min$^{-1}$ µg$^{-1}$ for the OP$^{DTT}$ and from $-0.018 \pm 0.152$ nmol min$^{-1}$ µg$^{-1}$ to
$0.197 \pm 0.104$ nmol min$^{-1}$ µg$^{-1}$ for the OP$^{AA}$. Such results agree with previous studies reporting different reactivity (or intrinsic OP) for different sources based on receptor-model techniques (Ayres et al., 2008; Bates et al., 2015; Cesari et al., 2019; Costabile et al., 2019; Fang et al., 2016; Paraskevopoulou et al., 2019; Perrone et al., 2019; Verma et al., 2014; Weber et al., 2018; Zhou et al., 2019; Daellenbach et al., 2020).

### 3.3.4 Intrinsic OP$^{DTT}$ and OP$^{AA}$

The road traffic source is the most reactive source towards OP$^{DTT}$, with a value of about $0.223 \pm 0.085$ nmol min$^{-1}$ µg$^{-1}$. Such value is twice higher than the ones observed for other significantly reactive sources, namely MSA-rich, biomass burning, dust and primary biogenic sources with OP$^{DTT}$ values of about $0.132 \pm 0.410$ nmol min$^{-1}$ µg$^{-1}$, $0.129 \pm 0.065$ nmol min$^{-1}$ µg$^{-1}$, $0.121 \pm 0.114$ nmol min$^{-1}$ µg$^{-1}$ and $0.112 \pm 0.113$ nmol min$^{-1}$ µg$^{-1}$, respectively. Interestingly, the nitrate-rich factor is well correlated to the OP$_{v}^{DTT}$ (r=0.43) but associated to the lowest intrinsic OP$^{DTT}$ value ($0.044 \pm 0.065$ nmol min$^{-1}$ µg$^{-1}$). Overall,
the OP$^{DTT}$ is sensitive to more sources than the OP$^{AA}$ as already pointed by Fang et al. (2016) and Weber et al. (2018), and seems to target all the sources containing either metals and organic species. However, it is not sensitive to the nitrate-rich source.

Based on the intrinsic OP$^{AA}$ results, a small number of PM sources shows significant redox activity, as already pointed out in previous studies (Bates et al., 2019, and references therein). Only the biomass burning, and road traffic sources show intrinsic
OP values significantly higher than 0 ($0.197 \pm 0.103$ nmol min$^{-1}$ µg$^{-1}$ and $0.161 \pm 0.108$ nmol min$^{-1}$ µg$^{-1}$, respectively).

We then confirm what previous studies found for these two assays, either by direct OP measurements at the emission source or by source apportionment. It is however hard to directly compare the absolute values from our results to the literature since the measurement protocols used are highly variable from one study to another.

### 3.3.5 Variability of the intrinsic OP's

The coefficient of variation (CV, standard deviation over mean) of the intrinsic OP's are the lowest for the **biomass burning** and **primary road traffic** for the DTT assay with values of 0.50 and 0.38, respectively, as well as for the AA assay with value of 0.52 and 0.67, respectively.





### 3.3.6 Biomass burning

The variability of the biomass burning intrinsic OP is somewhat site dependent, with a low uncertainty within a given site, but
with slightly different intrinsic OP between sites. It suggests that the variability is not linked to uncertainties of the model but
may be due to actual local variations of the chemical composition of this profile.

This result contrasts with the fact that the biomass burning was identified as a stable profile, with a PD < 0.1 and SID < 0.7
(Figure S1). Hence, the variability of OP's intrinsic values may come from species not directly measured in our dataset. Namely,
no polycyclic aromatic hydrocarbons (PAHs), oxy-PAHs, OH-PAHs, nitro-PAHs and especially polyaromatic quinones were
measured, although they are known to generate ROS and to contribute to the OP (Bolton et al., 2000; Charrier and Anastasio,
2012; Chung et al., 2006; McWhinney et al., 2013; Lakey et al., 2016; Jiang et al., 2016; Tuet et al., 2019; Gao et al., 2020;
Bates et al., 2019) and have short live time and being heavily influenced by the climatic condition (Miersch et al., 2019). It is
the same for the HULIS component of the biomass burning emissions.

### 3.3.7 Road traffic

Contrasting with the biomass burning factor, the uncertainty of the road traffic intrinsic OP at each site lies in the uncertainties
of the other sites. Hence, the low variability for the $OP^{DTT}$ indicates that the main components of the road traffic chemical
profiles may be the ones that primarily influence $OP^{DTT}$. However, for the $OP^{AA}$ the variability is higher with some important
differences from site to site, without clear distinction by typology or groups of sites. Then, some chemical species that are
not measured here may influence the $OP^{AA}$ with their variabilities in concentrations, but not the $OP^{DTT}$. These un-measured
species might also result from the variable contributions at each site of the different traffic-related sources, notably the extend
of SOA, or that of exhaust and non-exhaust emissions, that could have high impact on OP, as shown by Daellenbach et al.
(2020).

It is however interesting to note that the species known to contribute to the oxidative potential have low concentration
variability in this factor. Notably, the Copper, which is suspected to play a key role in the observed $OP^{DTT}$ and, moreover,
to the $OP^{AA}$, has rather low concentration uncertainties into the road traffic chemical profile (see http://getopstandop.u-ga.fr/
results?component=pmf_unc), as well as its concentration variation across sites. Indeed, Cu is largely apportioned by the road
traffic source (between 34% to 54% (first and third quartile, http://getopstandop.u-ga.fr/results?component=pmf_profiles), with
a concentration ranging from $1.7\,\mathrm{ng\,\mu g^{-1}}$ to $3.1\,\mathrm{ng\,\mu g^{-1}}$ of PM from this source.

### 3.3.8 Secondary inorganic factors (nitrate- and sulfate-rich)

The inorganic factors (sulfate-rich and nitrate-rich) present high CV's for their intrinsic OP values. However, the CV might not
be an accurate measure for some sources with near-zero mean intrinsic OP. The standard deviations are similar to the one of
the biomass burning and road traffic for the $OP^{DTT}$ and are among the lowest variability for the $OP^{AA}$. Both inorganic factors
are also very similar at each site in term of chemical composition, as presented in the SID-PD space in Figure S1. We then





confirm previous analysis on the low impact of secondary inorganic aerosols (SIA) on OP measurement (Daellenbach et al.,
2020) in favor of the subsequent moderate role of SIA for human toxicity (Cassee et al., 2013).

### 3.3.9   Mineral dust

The dust source presents an important variability when considering all series, but a deeper analysis showed 2 groups of sites:
AIX-RBX-VIF vs all other sites. The first group presents high intrinsic OP for both assays, whereas sites from the second group
display halved ($OP^{DTT}$) or almost null ($OP^{AA}$) intrinsic OP's. The first conclusion is that 80% of the sites presents a common
intrinsic OP for the dust source. The high variability observed in VIF may be explained by different chemical composition.
Indeed, the dust factor at VIF highly differs from the other dust factors with a PD > 0.75 when compared to other sites. We do
not have a clear hypothesis yet for the two other sites, but here again un-measured trace species might be responsible of these
differences, possibly coming from road dust resuspension and/or secondary processes leading to oxy-PAHs (Ringuet et al.,
2012a, b) or HULIS (Srivastava et al., 2018).

### 3.3.10   Secondary organics aerosols

It has been shown that, in general, biogenic SOA species may contribute to the OPDTT or to the generation of ROS (Jiang
et al., 2016; Tuet et al., 2017b; Park et al., 2018; Kramer et al., 2016; Manfrin et al., 2019). In our study, the MSA-rich factor
is the only one strictly included in this category. However, the intrinsic OP's of the MSA-rich source presents high variability
between sites with a CV of 3.1 and 7.8 for the DTT and AA assays, respectively, all sites combined, a variability also observed
for the values within each site. This secondary organic source appears to be the most variable source in term of intrinsic OP,
notably for the DTT assay.

This MSA-rich factor is also the lowest contributor to the PM mass of the 8 major sources, and the PMF bootstrap result
presents important variability for the fraction of $PM_{10}$ apportioned by this source. Further, the chemical profile is also quite
variable. Indeed, this PMF factor is not well described in the literature, and few studies reported it so far (Srivastava et al.,
2019; Lanzafame et al., 2020; Borlaza et al., 2020). As a result, we do not know for instance the loading of HULIS, quinone
or isoprene-derived-compounds contained in this factor, nor the amount of ageing it presents at each site. Hence, these uncer-
tainties on the additional chemical compounds included in this factor, despite the excellent tracer capability of the MSA itself,
might explain the diversity of its observed intrinsic OP's.

To a lesser extent, the sulfate-rich and aged sea-salt factors are also suspected to account for some SOA due to some amounts
of OC in their chemical profiles (around 2.5% of the total OC for both of them). It should be mentioned that, when adding
secondary organic tracers like 3-MBTCA to a set of PMF input data, the sulfate-rich source largely split for a secondary
biogenic factor, indicating such mixing if the proper tracers are not available (Borlaza et al., 2020). Without these tracers, such
mixing in the present study might explain a fraction of the variability of the intrinsic OP, at least for the sulfate rich-factor.





### 3.3.11 Primary biogenic

The primary biogenic source, mainly traced by polyols, presents some variability for the $OP^{DTT}$. Samaké et al. (2017) highlighted that spore or bacteria does contribute to the $OP^{DTT}$ and $OP^{AA}$ activities, even when the microbial cells are dead. However, the authors also present the inhibition of the DTT loss rate in presence of 1,4-naphtoquinone or Cu. The presence of both synergistic and antagonistic effects between species and microbiota might explain the variability of intrinsic $OP^{DTT}$ observed in Figure 3, reflecting the different local microbiology carried by the PM, or covariations of the primary biogenic
source with other metals or quinone rich sources for instance.

Another hypothesis to explain the variability of the intrinsic OP's might be the "ageing" of this factor, since Samaké et al. (2019b) pointed out that some secondary species may be incorporated in this factor at some sites, making it a mix of primary biogenic and SOA. It is then possible that the SOA mixed in the primary biogenic influenced the intrinsic OP in different ways, similarly to our hypothesis for the MSA-rich factor.

### 3.3.12 Overall geochemical agreement

Finally, despite the different PMF solution (and therefore the slightly different number of sources) obtained at each site and the different OP's signals, the rather low variability of the intrinsic OP determined for a given source suggests that most of the sources of PM behave similarly with regards to the OP over large geographical area in France. It then supports the idea that, at the national scale, the sources described above have a stable intrinsic OP, but that more precise values may be obtained when
using an even better PMF approach including some other more specific tracers.

### 3.4 Contribution of the sources to the OP's

The relative importance of the contributions of the sources to the total $PM_{10}$ OP's is weighted by their different intrinsic OP's. The question, therefore, is to what extend the contribution of the sources to the OP's differs from their contributions to the PM mass concentration. In this part, we present an aggregated view of the seasonal contribution of the sources to the OP in
Figure 4, and the daily contribution in Figure 5 and 6, considering all sites. Details per sites are presented in the associated website.

We would like to stress here that our dataset included an important proportion of alpine sites as well as urbanized sites. Then, the extrapolation to the whole France or to other regions of western Europe should be done cautiously.

### 3.4.1 Seasonality of the contribution by mass or OP's

As already shown by previous study in France (Petit et al., 2019; Srivastava et al., 2018; Waked et al., 2014; Weber et al., 2019), the seasonal mean contributions to the $PM_{10}$ mass show the importance of the biomass burning source, followed by the secondary inorganic (sulfate-rich and nitrate-rich), the dust and road traffic. As a direct consequence of the different intrinsic OP's for these sources, we do observe a redistribution of their relative importance for the total OP. Namely, the nitrate-rich source that may contribute to a significant amount to the $PM_{10}$ mass, notably in spring, barely contribute to the $OP_v^{DTT}$ nor to





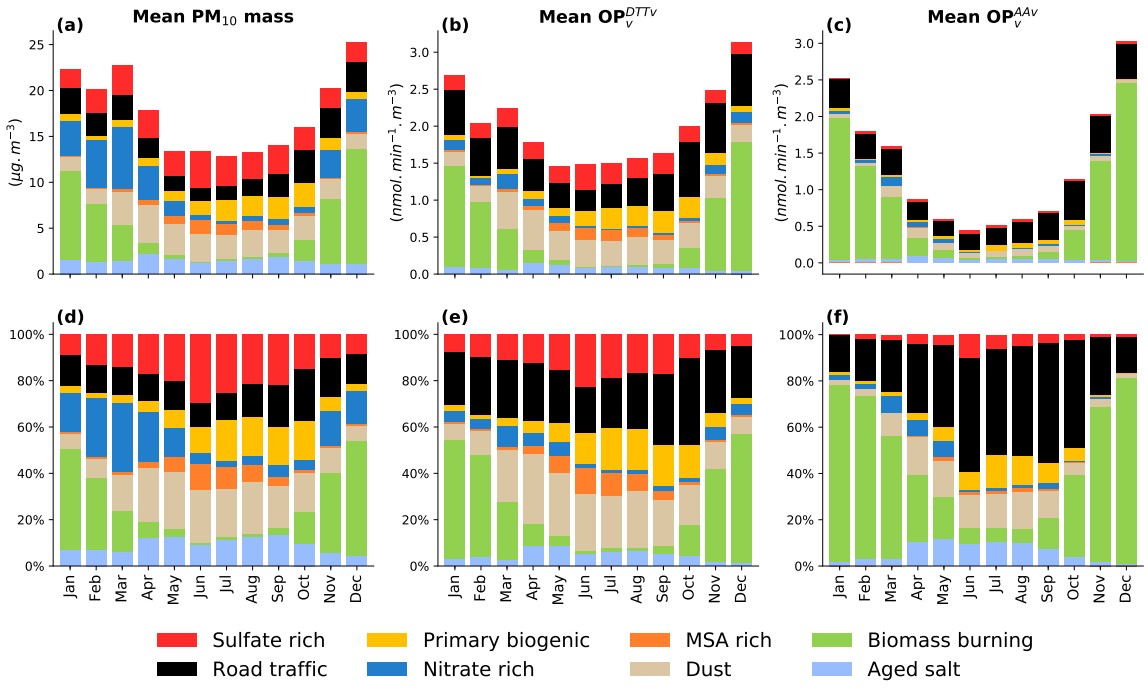

**Figure 4.** Mean monthly contribution of the main 8 sources to the **(a)** $PM_{10}$ mass, **(b)** $OP_v^{DTT}$ and **(c)** $OP_v^{AA}$ taken into account each sources contribution of every sites; their respective normalized contributions are presented in **(d)** $PM_{10}$ mass, **(e)** $OP_v^{DTT}$ and **(f)** $OP_v^{AA}$.

the $OP_v^{AA}$, in agreement with Daellenbach et al. (2020). Conversely, the road traffic contributes to about 15% during summer to the mean $PM_{10}$ mass but represents more than 50% of the mean $OP_v^{AA}$ in the same period (Figure 4 **(d)**, **(e)** and **(f)**). However, the biomass burning sources is still a major contributor to both the $OP_v^{DTT}$ and $OP_v^{AA}$ during the winter months. We note that the primary biogenic source also contributes to the $OP_v^{DTT}$ but to a lesser extent to the $OP_v^{AA}$. Finally, the dust source is an important contributor to the $OP_v^{DTT}$ but not to the $OP_v^{AA}$. These results confirm and extend what previous studies already found (Cesari

et al., 2019; Daellenbach et al., 2020; Weber et al., 2018). Overall, the main contributors to OP's are the three factors suspected to included anthropogenic SOA (biomass burning, road traffic and dust, including possible resuspension of road dust for the latter one). It follows that considering the seasonality of OP's, regulations should target the biomass burning emission in order to decrease the $PM_{10}$ OP's during winter by a large amount, but also the road traffic that contributes homogeneously to both OP around the whole year.

### 3.4.2 Daily mean and median contribution : insights for exposure

A concise view of the same results, this time on the daily-aggregated basis, is provided in Figure 5 and Figure 6, presenting the contributors to $PM_{10}$ mass and OP's, ranked in decreasing order. Figure 5 reports the typical "mean" daily value, a parameter generally used in the atmospheric community while Figure 6 reports the "median" daily value, often used in epidemiological





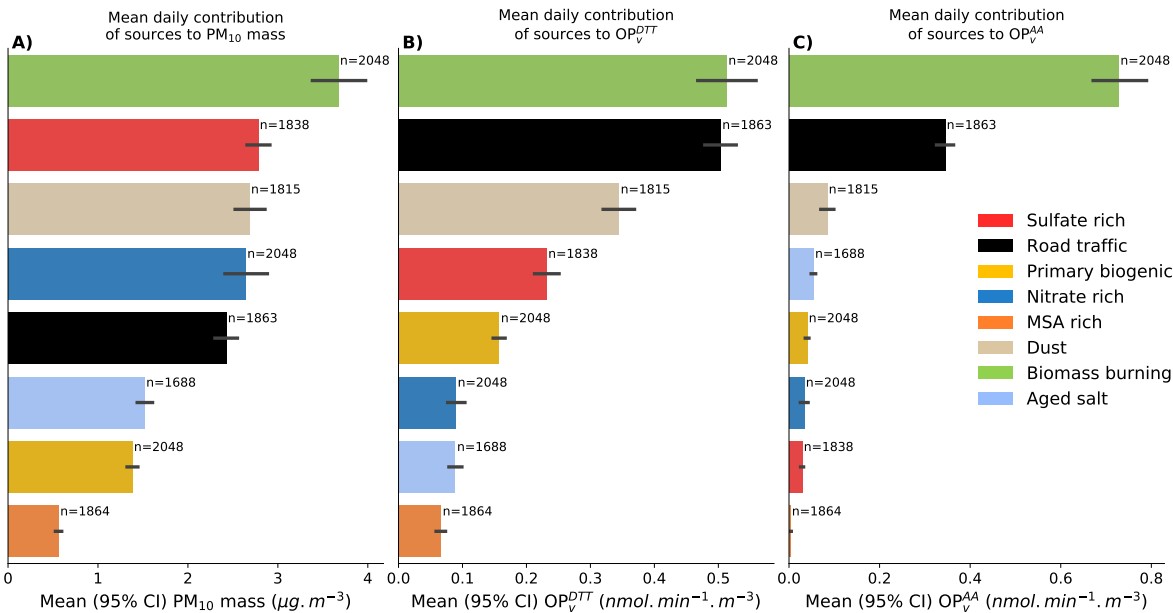

**Figure 5.** Averaged daily contribution of the sources to **(A)** the $PM_{10}$ mass, **(B)** the $OP_v^{DTT}$ and **(C)** the $OP_v^{AA}$. The bars represent the mean and the error bars the 95% confidence interval of the mean.

studies in order to discuss the chronical exposure of the population and avoid the high impact of unfrequent low or high events

that highly influence the mean value.

Due to the non-normality of the contribution, the results highly differ if considering the mean or the median contributions, and the two statistical indicators may not address the same questions. The skewness of the distribution is not surprising as some high $PM_{10}$ events (i.e., short time event in the dataset) were present in our measurements. This is also specifically anticipated in alpine areas (CHAM, PAS, MNZ, VIF, GRE-cb, GRE-fr_2013 and GRE-fr_2017) where the frequent development of

atmospheric thermal inversions layers in winter is causing increased pollutant concentrations.

We observed in Figure 5 a redistribution of the daily **mean** contribution sources' rank between the $PM_{10}$ mass, $OP_v^{DTT}$ and $OP_v^{AA}$ similarly to the monthly mean contribution discussed above. The biomass burning source being an important contributor to the $PM_{10}$ mass, contributes also significantly to both OP and is ranked as the first contributor to both $OP_v^{DTT}$ and $OP_v^{AA}$ mean daily contribution (mean $0.51 \, \mathrm{nmol \, min^{-1} \, m^{-3}}$ and $0.72 \, \mathrm{nmol \, min^{-1} \, m^{-3}}$, respectively). The road traffic source contribution,

due to its highest intrinsic OP in both assays, presents almost the same daily mean contribution than the biomass burning for the $OP_v^{DTT}$, and is the second contributor to the daily mean $OP_v^{AA}$, with half the contribution of the biomass burning (mean $0.50 \, \mathrm{nmol \, min^{-1} \, m^{-3}}$ and $0.34 \, \mathrm{nmol \, min^{-1} \, m^{-3}}$, respectively). The other sources barely contribute to the $OP_v^{AA}$ ($<0.1 \, \mathrm{nmol \, min^{-1} \, m^{-3}}$). For the $OP_v^{DTT}$, the dust is the third contributor (mean $0.34 \, \mathrm{nmol \, min^{-1} \, m^{-3}}$), followed by the sulfate-rich and primary biogenic ($0.23 \, \mathrm{nmol \, min^{-1} \, m^{-3}}$ and $0.16 \, \mathrm{nmol \, min^{-1} \, m^{-3}}$, respectively). The nitrate-rich, aged





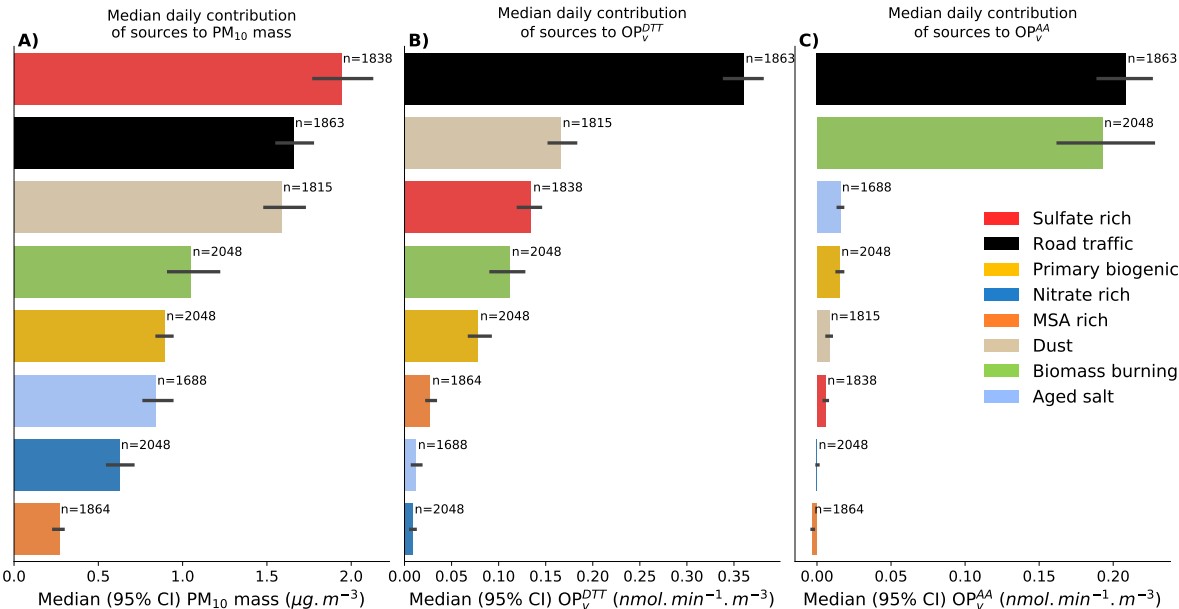

**Figure 6.** Median daily contribution of the sources to (**A**) the PM mass, (**B**) the $OP_v^{DTT}$ and (**C**) the $OP_v^{AA}$. The bars represent the mean and the error bars the 95% confidence interval of the median.

seasalt and MSA-rich present a low contribution (mean $<0.1 \, \mathrm{nmol \, min^{-1} \, m^{-3}}$), due either to their low intrinsic OP or to their low contribution to the PM mass.

    However, for the daily **median** contribution, due to the high seasonality of the biomass burning source and the consistent contribution throughout the year of the primary road traffic, sulfate-rich and dust sources, the ranks of the sources are drastically redistributed between the 3 metrics considered (Figure 6). Moreover, the absolute values of the contributions are also lowered
compared to the mean daily contribution, due to low frequencies of highly loaded PM events. The major source contributing to the $OP_v^{DTT}$ is now the primary road traffic (median $0.36 \, \mathrm{nmol \, min^{-1} \, m^{-3}}$), contributing more than twice as much as the second source, namely the dust one (median $0.16 \, \mathrm{nmol \, min^{-1} \, m^{-3}}$), followed by the sulfate-rich (median $0.13 \, \mathrm{nmol \, min^{-1} \, m^{-3}}$) and then the biomass burning (median $0.11 \, \mathrm{nmol \, min^{-1} \, m^{-3}}$). For the $OP_v^{AA}$, the two dominant sources are the primary road-traffic (median $0.29 \, \mathrm{nmol \, min^{-1} \, m^{-3}}$) and the biomass burning (median $0.19 \, \mathrm{nmol \, min^{-1} \, m^{-3}}$), all other sources being now
negligible (aged salt $0.016 \, \mathrm{nmol \, min^{-1} \, m^{-3}}$, primary biogenic $0.015 \, \mathrm{nmol \, min^{-1} \, m^{-3}}$ and the others contributes less than $0.01 \, \mathrm{nmol \, min^{-1} \, m^{-3}}$).

    The high differences between the mean and median contributions could have strong implication for air quality policies. Indeed, as previously shown, the biomass burning may contribute to more than 50% of the high OP's during winter, and even more for some days. However, such events do not well represent the daily exposure of the population over the full year. Even if
the regulations should target those events to prevent acute exposure, they should also strongly take into account the long-term exposure to a lower but constant level of pollutant, since there is no threshold below which PM are no longer harmful (WHO,





2013). With this respect, the emissions from the road traffic becomes a major concern as well, supporting that this source might actually be the most important one to be targeted in order to decrease the chronic exposure to PM pollutant.

## 3.5 Results of OP's inversion for local PMF factors

Some other factors were obtained in a limited set of PMF inversions, and their chemical profiles are somewhat variable. It is however interesting to discuss of their impact on their OP's values.

### 3.5.1 Heavy fuel oil

The Heavy Fuel Oil (HFO) source is identified at MRS-5av and PdB, both sites being large port on the Mediterranean coast. It presents an intrinsic $OP^{DTT}$ of $0.51 \pm 0.14 \, \text{nmol} \, \text{min}^{-1} \, \mu\text{g}^{-1}$ and $0.21 \pm 0.04 \, \text{nmol} \, \text{min}^{-1} \, \mu\text{g}^{-1}$, respectively, and an intrinsic 465 $OP^{AA}$ of $0.04 \pm 0.02 \, \text{nmol} \, \text{min}^{-1} \, \mu\text{g}^{-1}$ and $0.11 \pm 0.03 \, \text{nmol} \, \text{min}^{-1} \, \mu\text{g}^{-1}$, respectively. The intrinsic $OP^{DTT}$ is then in average higher than the road traffic one, making HFO the second contributor of the daily mean and median source contribution at MRS-5av for the $OP^{DTT}$ contribution and the fourth one for the $OP^{AA}$ contribution (see website). For the PdB site, the contributions are a bit lower. Although only 2 sites presented an HFO factor, and similarly to previous studies (Hu et al., 2008; Kuang et al., 2017; Moldanová et al., 2013; Wang et al., 2020), it suggests that the PM originated from this source may significantly 470 contribute to the total OP around harbor cities.

### 3.5.2 Industrial

As already stated, the chemical composition of this profile highly varies from site to site, for the 6 sites where it is determined. Therefore, the intrinsic OP's of this profile are also highly variable. The intrinsic $OP^{DTT}$ are high for GRE-cb and GRE-fr_2017 ($0.52 \pm 0.30 \, \text{nmol} \, \text{min}^{-1} \, \mu\text{g}^{-1}$ and $0.37 \pm 0.27 \, \text{nmol} \, \text{min}^{-1} \, \mu\text{g}^{-1}$, respectively), as are the intrinsic $OP^{AA}$ ($0.82 \pm$ 475 $0.29 \, \text{nmol} \, \text{min}^{-1} \, \mu\text{g}^{-1}$ and $0.61 \pm 0.17 \, \text{nmol} \, \text{min}^{-1} \, \mu\text{g}^{-1}$, respectively). However, both are close to 0 for the other sites where this factor is found (PdB, AIX, TAL and VIF). The high intrinsic OP seems to indicate again the role of metals in the OP of PM, however since this factor has strong uncertainties associated with the PMF results, and then to the intrinsic OP, further work on the source profiles is needed to draw firmer conclusion.

## 4 Limitations of the study

In this study, we focused on major sources and trends, hence limit our study to some aspects. Notably the PMF standardized approach allows common source identification at the national scale but may also dampers some site specificities. Also, the choice to focus on the main sources of PM and to discuss the aggregated results shorten the discussion on some local specificity, notably potential local sources that contribute to the OP (for instance HFO or industry), that may be relevant for some sites but is not applicable to a wider area.

One main limitation is also the use of linear regression tools whereas it has been shown that OP is not fully proportional to the mass of compounds. The residual analysis seems to agree with this experimental finding since the highest OP samples is



underestimated by the MLR model. The addition of co-variation term or even the use of non-linear regression may be the next step to better explain the OP of the sources (Borlaza et al.).

Moreover, if the intrinsic OP results from the MLR can be extrapolated to any given site with similar regional background
of the urbanized area used in this study, the source contributions extrapolation should be taken cautiously since our dataset displays an over-representation of the alpine sites with regard to the whole France.

## 5    Conclusions

To our knowledge, this study gathers the most important database of OP samples, with concomitant observations of chemistry analysis, source-apportionment through PMF, and the measure of two OP assays (DTT and AA) for 15 yearly time series over
France spanning between 2013 to 2018 for a total of >1700 samples.

We demonstrated that source apportionment of OP through a "simple" multi linear regression without any constraint on the coefficient provides good statistical results and can explain the observed $OP_v^{DTT}$ and $OP_v^{AA}$.

- – The intrinsic OP's of the main regional sources present values in the same range at each site, especially for the primary road traffic, biomass burning, nitrate-rich, dust, and sulfate-rich PMF factors. Biogenic and MSA-rich factors present
higher discrepancy according to the site together with the highest uncertainties at each site.

- – Some site-specific sources might have an important intrinsic OP and can account for a non-negligible part to the observed OP (notably in harbor cities or near industrial site).

- – Different sensitivities for the two OP assays towards a given source are highlighted. The DTT appears to be sensitive to a wide range of sources, whereas the AA targets mainly the biomass burning and primary road traffic factor.

- – With consistency at the regional scale, the primary road traffic and biomass burning factor are the main absolute OP contributors, together with dust for $OP^{DTT}$ to a lesser extent. Conversely, the secondary inorganic sources (nitrate- and sulfate-rich) barely contribute to OP's.

- – In order to assess the chronic population exposure, the median daily contribution of sources to the $OP_v^{DTT}$ and $OP_v^{AA}$ are also reported and present important differences in ranking compared to the mean contributions. The importance of
the primary road traffic source drastically increases, notably for the $OP_v^{DTT}$, whereas the biomass burning contribution is lowered. However, only the road traffic and biomass burning sources contribute to the daily median of the $OP_v^{AA}$.

Finally, the relatively stable intrinsic OP at a large geographical scale for the main PM sources allows future work on to the implementation of the OP into regional chemistry transport model. This step would allow a quantitative estimation of the population exposure OP, expending potential cross-over studies with epidemiology.

*Data availability.*   Available on request





*Code and data availability.* Available on request

## Appendix A: Correlation between OP, chemical species, and sources

The Spearman correlations between the chemical species and $OP_v^{DTT}$ and $OP_v^{AA}$ are presented in Figure A1, while the Spearman correlations between the source mass apportionment from the PMF and the measured $OP_v^{DTT}$ and $OP_v^{AA}$ are presented in Figure A2. All samples from all sites were considered in the results presented here.

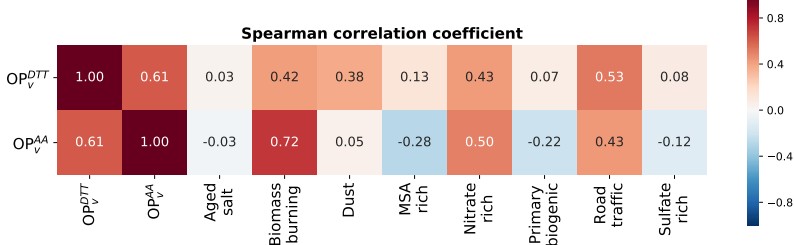

**Figure A1.** Spearman correlation coefficients between $OP_v^{AA}$ and $OP_v^{DTT}$ and the chemical species observed at each site. The numbers of samples are specified into brackets.

First, we can see that both OP assays correlated well with each other ($r_{OP\ DTT-OPAA}$=0.61) but do not present an exact similitude. Notably, the $OP_v^{AA}$ has stronger seasonality than $OP_v^{DTT}$ and higher correlation are found during winter than summer. Details of the individual timeseries are given in the website.

Second, the only source that strongly correlates to one OP (r>0.6) is the biomass burning to the $OP_v^{AA}$. Some low to mild correlation (0.3<r<0.6) are found for the $OP_v^{DTT}$ vs. road traffic, biomass burning, nitrate-rich and dust and for the $OP_v^{AA}$ vs. nitrate rich and road traffic. The copper, mostly apportioned by the road traffic, is the most correlated metals to the $OP_v^{AA}$ and the second for $OP_v^{DTT}$. For the levoglucosan and mannosan, strong correlations are found to the $OP^{AA}$. These results are in agreement with previous studies, either with the source correlation (Weber et al., 2018) or with the proxy of sources (namely, levoglucosan for biomass burning and EC, iron, copper or PAH for road traffic) (Figure A1 and Calas et al. (2018, 2019); Charrier and Anastasio (2012, 2015); Cho et al. (2005); Hu et al. (2008); Janssen et al. (2015); Künzli et al. (2006); Ntziachristos et al. (2007); Pietrogrande et al. (2018); Verma et al. (2009, 2014, 2015a); Borlaza et al. (2018).

Figures A1 and A2 also indicate that the nitrate and nitrate-rich source concentrations mildly correlate to both OP. However, neither nitrate not the chemical species included in the chemistry profile of the nitrate-rich source present redox-active capabilities. Conversely, crustal elements (mainly Ti and $Ca^{2+}$) present none to low correlation to OP's. Although mineral dust has been reported to contribute to the OP, the episodic event of Saharan wind may be insufficient to be reflected in the simple $r^2$ value. Also, as already stated, the strong seasonal cycle leads to negative correlation between some sources or species and the OP (MSA or polyols species, and MSA-rich and primary biogenic factors). When considering only the warm period, they





appear positively correlated to both OP's due to the exclusion of the strong impact of the biomass burning source (not shown here).

These examples indicate that it is hard to assess robustly the links between OP's and chemistry if using only correlations. Moreover, even a good correlation may not reflect any causality and the multilinear regression should disentangle possible co-variation due to meteorological effect (accumulation or long-range transport).

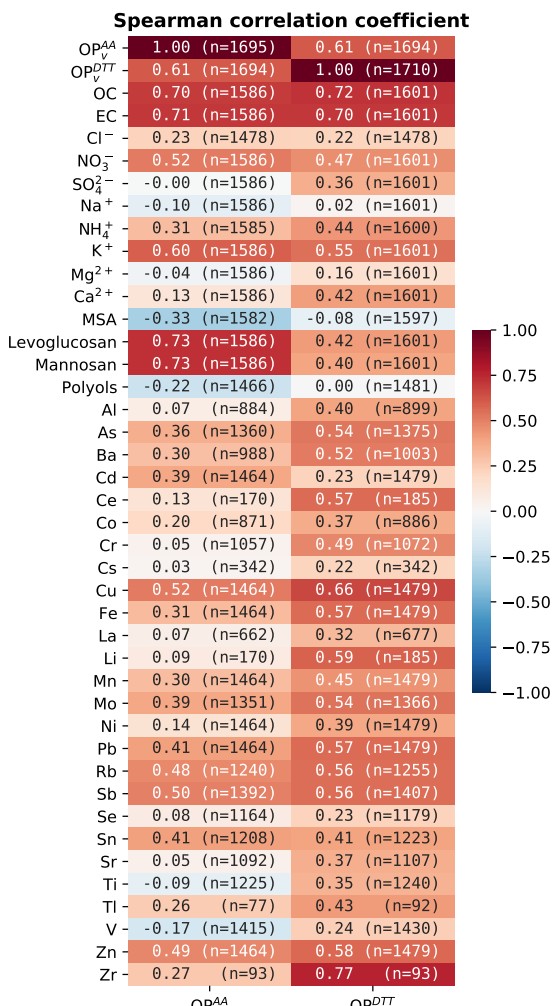

**Figure A2.** Searman correlation coefficient between $OP_v^{DTT}$ or $OP_v^{AA}$ and the different PM sources identified for at least two third of the sites. All sites are merged and whole time-series of measurements were considered. The number of observations considered for the different sources are as follows: Aged salt: 1430; Biomass burning: 1700, Dust: 1489, MSA rich: 1595, Nitrate rich: 1700, Primary biogenic: 1700, Road traffic: 1587, Sulfate rich: 1524.



*Author contributions.* O.F, J.-L. J, G.U and J-L. B were in charge of the coordination of different research programs and funding acquisitions. D.S did the data curation and ran the PMF for the SOURCES program, F.C. and J.A. did the data curation and ran the PMF for

the DECOMBIO program, S.W. and L.B did the data curation and ran the PMF for the Mobil'Air program. A.C and G.U set up the 2 OP assay methodologies. S.W designed the methodology, did the formal analysis and prepared the present manuscript and figures. GU and JLJ designed, reviewed and edited the first draft of the manuscript. All the co-authors read and edited the manuscript.

*Competing interests.* The authors declare no competing interests.

*Acknowledgements.* This work was partially funded by ANSES for OP measurements (ExPOSURE program, grant 2016-CRD-31), IDEX UGA grant for innovation 2017 ROS-ONLINE and CDP IDEX UGA MOBILAIR (ANR-15-IDEX-02). It was also supported by the French Ministry of Environment, as part of the national reference laboratory for air quality monitoring (LCSQA, program CARA), for some of the chemical analysis related to GRE-fr, TAL, RBX, STG-cle, MRS-5av and NIC stations. The study in CHAM, MNZ and PAS was funded by ADEME and PRIMEQUAL within the DECOMBIO program (1362C0028). The studies in GRE-cb and VIF were funded by the UGA IDEX

Mobil'Air program (ANR-19-CE34-0002-01) and the Ademe QAMECS program (1262c0011). Regional monitoring networks (namely, Atmo AuRA, Atmo Sud, Atmo HdF, Atmo NA and Atmo GE) financially contributed to sampling and/or chemical analyses for samples from their respective sites. The PhD of Samuël Weber was funded by a grant from ENS Paris. This study was also supported by direct funding by IGE (technician salary), the LEFE CHAT (program 863353: "Le PO comme proxy de l'impact sanitaire"), and LABEX OSUG@2020 (ANR-10-LABX-56) (for funding analytical instruments).

The authors wish to thank all the numerous people (who couldn't be listed exhaustively here) from the different laboratories (IGE and Air-O-Sol analytical platform, Ineris and PTAL analytical platform of EDYTEM) and from regional air quality monitoring networks listed as co-authors' affiliations who actively concurred in filter sampling and/or analysis. The Institute Mines Telecom (IMT-Lille Douai, PI: L.Y. Alleman) and the Laboratoire des Sciences du Climat et de l'Environnement (LSCE, PI : N. Bonnaire) are also warmly acknowledged for their contribution to chemical analysis of samples collected at RBX, NGT TAL and STG-Cle.



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
