# Peer review of "Source apportionment of atmospheric $PM_{10}$ Oxidative Potential: synthesis of 15 year-round urban datasets in France"

_Atmospheric Chemistry and Physics, 2021_

## Author Comment (AC1)

**Answer to referee#1**

This manuscript is focused on the source apportionment of the oxidative potential (OP) of atmospheric $PM_{10}$ from a long-term large-scale sampling campaign in France. More than 1700 samples from 14 sites and 15 years sampling period were analyzed for OP in two endpoints ($OP^{AA}$ and $OP^{DTT}$) and various chemical species. The authors implemented a positive matrix factorization (PMF) coupled with multiple linear regression (MLR) protocol for identifying the sources from individual sites contributed to measured OP. The importance of different sources was discussed based on both intrinsic OP (OPm) and their contributions to the total OPv. The authors also attempted to explain the seasonality of certain sources in the context of OP. Overall, the manuscript presented very interesting and significant results of $PM_{10}$ OP in West Europe area, and strongly supported the health effect study and legislation of air quality control in France. However, it lacks a discussion on OP's spatial distribution and the linkages of chemical composition to OP. Following are my specific comments.

We are thankful to the referee for their constructive feedback and the different comments and questions raised. Hereafter you will find our answer in blue.

Specific comments:

1. Generally, fine particles ($PM_5$) have been linked to the adverse effect caused to human respiratory and cardiovascular system, since these particles have highest penetration efficiency in lower respiratory tract. However, the PM samples collected in this manuscript have involved larger sized particles which have shown lower oxidative potential (Hu et al., 2008;Ntziachristos et al., 2007) and somewhat lesser relevance to human health. Thus, a justification should be provided for using $PM_{10}$ in this health-related study, i.e. source apportionment of PM OP.

We, indeed, agree with the reviewer that a difference exists between $PM_{10}$ and $PM_{2.5}$, both in terms of processes and of particle size prone to deposition in the lungs. However, studies also point out the role of the coarse fraction of PM for health impact (Keet et al. 2018 ; Chen et al. 2019 ; Wang et al. 2018). Finally, in EU and France, daily limit values are set for $PM_{10}$ only and used as alert tool for health issues. $PM_{10}$ need to be investigated with this respect.

It should also be noted that the 2.5 µm threshold commonly used to differentiated fine and coarse aerosols - is mainly based on technical considerations, but do not always reflect properly the bimodal size distribution actually observed in ambient air. Various studies, including those mentioned by the Reviewer, showed clear differences both for deposition of OP in respiratory tracts and in OP activity depending on the size of PM; but with size threshold varying between 1.18 and 3.2 µm, which can already make great differences in the composition of the PM compared to the $PM_{2.5}$ fraction. For instance, in the study by Fang et al (2017), $Cu^{2+}$ mass concentrations are equally distributed between $PM_{2.5}$ and $PM_{2.5-10}$ fractions.

However, we added this remark as a limitation and edited the text as follows in the method section:

Even if it has been shown that mainly $PM_{2.5}$ deposit in lung alveoli (Fang et al., 2017), $PM_{10}$ are still a public health concern and under regulation in EU and France (Directive 2008/50/CE). Moreover, recent studies also highlight the role of the coarse fraction of PM for heatlh impact (Keet et al. 2018, Chen et al. 2019; Wang et al. 2018). $PM_{10}$ has the advantage to encompass all parts of PM potentially reaching the lower respiratory track.

2. Lines 85 – 89: Please provide more details on the geographical information of all sites, e.g. distance to local highways, industries and/or coast, wind direction and relative location in the valley/peak of Alpine area etc. This can help to directly identify some potential sources.

We agree that a detailed view of the site would help to better understand the different specificities of the sites. However, in this study, we do not want to focus on given sites. Instead, we want to elucidate common features, thanks to a large dataset, in order to extract the main impacts of sources to the OPs at a larger scale.

We extended the description of the Table S1 to give sufficient metadata information on the sites, classifying by typology (i.e Urban Background, Urban traffic (i.e. near an in-city highway), Traffic (near a main highway), Urban valley (surrounded by mountain) and Industrial (located in one of the most important France refinery area). Short paragraphs were also added in the supplementary information to picture each site in brief descriptions.

3. Line 119: Why 1,4-naphthoquinone was chosen as the positive control for AA? Does it have a consistent and high AA activity?

1,4-NQ is used as positive control due to its constant and predicted high OP. We use 25 µM 1,4NQ as positive control (6.6 µM final concentration in the well) and observe a constant two to three fold higher AA activity in comparison to ambient PM samples.

4. The apportionment of SOA factors in this paper is ambiguous. The authors did not explain the contributions of OC in three highlighted SOA-related factors, i.e. nitrate-rich SIA, sulfate-rich SIA and MSA-rich. We suggest a better explanation should be provided on their formation and the differences of their contributions to OP and mass of $PM_5$.

We agree with the reviewer that SOA fractions are very difficult to clearly identify and quantify in such source apportionment studies. Some developments were recently undertaken to address this issue, notably in France (e.g., Srivastava et al. 2018a, 2018b, 2019 ; Petit et al. 2019 ; Borlaza et al. 2021). However, such methodologies could not be used for the sites/datasets used in the present study, and we do not see the need for having an extensive discussion on his topic here, which would also add a lot of content to this manuscript. Such a discussion has already been proposed in previous papers (e.g., Waked et al., 2014; Weber et al., 2019 ; Favez et al, 2021). In particular, the details of the chemical profiles obtained for each of the PMF factors at each site used in the present study can be found at http://getopstandop.u-ga.fr/results?component=pmf_profiles and are discussed in Weber et al. (2019).

Thanks to this remark of the reviewer, we notice a typo in the manuscript where the OC apportioned by the **sulfate-rich** should read **nitrate-rich** factor line 375:

Previous sentence:

To a lesser extent, the sulfate-rich and aged sea-salt factors are also suspected to account for some SOA due to some amounts of OC in their chemical profiles (around 2.5% of the total OC for both of them).

In fact, the sulfate rich one accounts for around 14% of the total OC, with some important variability between sites (from 5 to 33%). Nevertheless, this full section has been eventually re-written, and now presents a deeper discussion on the chemical component of the different factors, also considering our answer to the next comment raised by the Reviewer.

5. The subsections in Section 3.3 are confusing. I suggest combining Sections 3.3.2 – 3.3.5 to a single subsection "Intrinsic OP of main PMF sources", and create a new subsection for Sections 3.3.6 – 3.3.12 (like Section 3.4), "Profiles of OPm sources".

We thank the reviewer for this suggestion. Due to a formatting issue, the heading level was missing. A similar concern was raised by reviewer #2, so we reworked this section to account for it. We also discussed more in depth the different factors here, following the comments of both reviewers.

6. Lines 367 – 370: The authors tend to tone down the intrinsic OP of MSA-rich factor. In fact, I found very high $OP^{DTT}m$ for this factor at GRE-cb, NGT and STG-cle with reasonable CV (<0.6), while $OP^{AA}m$ at these sites were near 0. I would suggest the authors to further explore the redox-active components and explore the reasons for different activities between two endpoints for this factor at these sites.

Indeed, the MSA-rich factor may be considered as associated with an important intrinsic OP in some sites, but the inter-site variability is very high. Moreover, this PMF factor is identify thanks to a single marker (MSA) and very few studies only have reported it so far. The exact primary sources or processes leading to this factor are still under discussion for non-arctic regions (Golly et al. 2018). We wanted to tone down the impact of MSA-rich factor mainly due to the important uncertainties and variabilities of the chemical component of the factor, and the inter-site variability of intrinsic OP. For instance, GRE-cb and VIF are within 15 kilometers with similar sampling periods, but present respectively the highest and lower intrinsic $OP^{DTT}$ for this factor. It is indeed of great interest that for the AA assays, however, the MSA-rich factor appears more similar at all sites.

For now, it is not clear if the uncertainties are mainly due to the sources' chemical component variabilities or to the inversion method used. The low amount of PM mass and its important uncertainties apportioned by the different PMF for this factor (between 0.7 to 5.5% of total $PM_{10}$ mass) could also be an explanation of the variability of the intrinsic OP of this factor. Hence, we prefer not to conclude to a clear effect of this source on the OP.

This discussion has been added in the manuscript.

7. Lines 391 – 395: Please specify the factors with low variability of OPm and explain why the other factors show large variability.

We reworked the section 3.4 and subsequent ones, to clarify this discussion. We now present the variability thanks to the interquartile range of intrinsic OP, and discuss the variability in the text.

8. Lines 410 – 415: the authors should highlight the high contribution of sulfate-rich SIA factor to OP$^{DTT}$ In comparison to its marginal contribution to OP$^{AA}$v, it is also important to find out the ROS-active components in this factor for the difference between two OP endpoints.

We reworked this part and a full paragraph is now dealing with each factor, including Sulfate-rich.

9. Lines 421 – 425: The explanations on the mean and median values of contribution of sources are unclear. The authors should explain the meaning of these two values and justify why using them under different circumstances.

We clarified it at the beginning of the corresponding paragraph in the revised manuscript as follows:

Figure 5 reports the typical "mean" daily value, a parameter generally used in the atmospheric community while Figure 6 reports the "median" daily value, often used in epidemiological studies in order to discuss the chronical exposure of the population and avoid the high impact of unfrequent low or high events that highly influence the mean value.

Due to the non-normality of the contribution and the high contributions of some sources at some site (for instance, the biomass burning source in alpine valley), the results highly differ if considering the mean or the median contributions, and the two statistical indicators may not address the same questions (the mean is more related to the identification of the major sources contributors and the median to the exposure of population).

10. Lines 473 – 476: I suggest the authors further investigate the sensitive chemicals (like transition metals) to both OP endpoints at all the sites with the industrial factor, in order to explain the huge difference of OPm found among all six sites.

The industrial factor determined is only determined at 6 sites by PMF and is one of the most heterogenous factor (see Figure S1). According to the uncertainties (both estimated by the bootstrap and displacement method of the EPA PMF5 software), the exact loading of metals is highly variable. Some of the statistical solution even led to unrealistic industrial factor with contributions from some metals that are higher than the total mass of PM apportioned.

The full chemical profiles (with uncertainties) are available to the readers at http://getopstandop.u-ga.fr/results?component=pmf_profiles, and clearly indicate a huge contribution of metals (notably, As, Cr, Mn, Mo, Ni, Pb, Rb and Zn)

Then, we think that a detailed discussion of the link between OP and chemicals for this factor would not bring much information since this factor represents very different types of sources. To be able to discussed it into more details, efforts should first be focused on having a clearer and more stable industrial factor at each of the corresponding site (e.g., using additional markers that we unfortunately didn't have here).

11. Line 507: A considerable contribution from sulfate-rich factor has been noted in Figure 4 – 6. Hence, the author made a wrong statement here saying both SIA sources contribute barely to OP which should be corrected.

The reviewer is indeed right. We should have been clearer between the difference of $OP^{DTT}$ and $OP^{AA}$. The sulfate-rich is still of importance with regard to the DTT, but not at all for the AA. We corrected the text and conclusion.

12. Line 512: The paper showed many sources with different spatial variability in intrinsic OP, which is not discussed. Therefore, caution should be exercised when making statements like "relatively stable intrinsic OP at a large geographical scale". The authors should discuss it in two types of sources – sources with low variability and stable intrinsic OP (e.g. road traffic), and sources with high variability and varied intrinsic OP (e.g. biomass burning).

We agree that we did not highlight enough the variabilities. Section 3.4 has been re-written with this comment in mind. We now set up an objective criterion to distinguish between a « variable » or « stable » intrinsic OP based on the interquartile value (Q25-Q75). We also stressed in the main text if the variability is related to spatial-variability (i.e. site presenting different intrinsic OP) or if it is due to important uncertainties for the results obtained by MLR model analyses.

Technical corrections :

1. Line 14: to prioritized – insert "be" between these two words.
2. Line 76: "multilinear" should be changed to "multiple linear" or "multivariate linear".
3. Line 115: the abbreviation in the parenthesis should be "RTLF".
4. Line 174, 237, 257: "specie" should be changed to "species".
5. Line 266, 459: The word "inversion" in the title is confusing. I suggest replacing "OP's inversion" with "intrinsic OP".
6. Line 431: I suggest moving "in Figure 5" to the end of the sentence or include it in parenthesis.
7. Legend of Figures 5 and 6: the error bars __ the 95% confidence level – insert "represent" at the underline.
8. Legend of Figure A2: "Searman" should be changed to "Spearman".

We thank the reviewer for its careful reading. Corrections were done accordingly in the revised manuscript.

**References**

Hu, S., Polidori, A., Arhami, M., Shafer, M., Schauer, J., Cho, A., and Sioutas, C.: Redox activity and chemical speciation of size fractioned PM in the communities of the Los Angeles-Long Beach harbor, Atmospheric Chemistry and Physics, 8, 6439-6451, 10.5194/acp-8-6439-2008, 2008.

Ntziachristos, L., Froines, J. R., Cho, A. K., and Sioutas, C.: Relationship between redox activity and chemical speciation of size-fractionated particulate matter, Particle and fibre toxicology, 4, 5, 2007.

**References**

Fang, T., Zeng, L., Gao, D., Verma, V., Stefaniak, A. B., and Weber, R. J.: Ambient Size Distributions and Lung Deposition of Aerosol Dithiothreitol-Measured Oxidative Potential: Contrast between Soluble and Insoluble Particles, Environ. Sci. Technol., 51, 6802–6811, https://doi.org/10.1021/acs.est.7b01536, 2017.

Wang, X., Qian, Z., Wang, X., Hong, H., Yang, Y., Xu, Y., Xu, X., Yao, Z., Zhang, L., Rolling, C. A., Schootman, M., Liu, T., Xiao, J., Li, X., Zeng, W., Ma, W., and Lin, H.: Estimating the acute effects of fine and coarse particle pollution on stroke mortality of in six Chinese subtropical cities, Environmental Pollution, 239, 812–817, https://doi.org/10.1016/j.envpol.2018.04.102, 2018.

Chen Renjie, Yin Peng, Meng Xia, Wang Lijun, Liu Cong, Niu Yue, Liu Yunning, Liu Jiangmei, Qi Jinlei, You Jinling, Kan Haidong, and Zhou Maigeng: Associations between Coarse Particulate Matter Air Pollution and Cause-Specific Mortality: A Nationwide Analysis in 272 Chinese Cities, Environmental Health Perspectives, 127, 017008, https://doi.org/10.1289/EHP2711, n.d.

Keet, C. A., Keller, J. P., and Peng, R. D.: Long-Term Coarse Particulate Matter Exposure Is Associated with Asthma among Children in Medicaid, Am J Respir Crit Care Med, 197, 737–746, https://doi.org/10.1164/rccm.201706-1267OC, 2018.

Srivastava, D., Favez, O., Perraudin, E., Villenave, E., and Albinet, A.: Comparison of Measurement-Based Methodologies to Apportion Secondary Organic Carbon (SOC) in $PM_{2.5}$: A Review of Recent Studies, 9, 452, https://doi.org/10.3390/atmos9110452, 2018a.

Srivastava, D., Tomaz, S., Favez, O., Lanzafame, G. M., Golly, B., Besombes, J.-L., Alleman, L. Y., Jaffrezo, J.-L., Jacob, V., Perraudin, E., Villenave, E., and Albinet, A.: Speciation of organic fraction does matter for source apportionment. Part 1: A one-year campaign in Grenoble (France), Science of The Total Environment, 624, 1598–1611, https://doi.org/10.1016/j.scitotenv.2017.12.135, 2018b.

Srivastava, D., Favez, O., Petit, J.-E., Zhang, Y., Sofowote, U. M., Hopke, P. K., Bonnaire, N., Perraudin, E., Gros, V., Villenave, E., and Albinet, A.: Speciation of organic fractions does matter for aerosol source apportionment. Part 3: Combining off-line and on-line measurements, Science of The Total Environment, 690, 944–955, https://doi.org/10.1016/j.scitotenv.2019.06.378, 2019.

Golly, B., Waked, A., Weber, S., Samake, A., Jacob, V., Conil, S., Rangognio, J., Chrétien, E., Vagnot, M.-P., Robic, P.-Y., Besombes, J.-L., and Jaffrezo, J.-L.: Organic markers and OC source apportionment for seasonal variations of $PM_{2.5}$ at 5 rural sites in France, Atmospheric Environment, 198, 142–157, https://doi.org/10.1016/j.atmosenv.2018.10.027, 2019.

Favez, O., Weber, S., Petit, J.-E., Alleman, L. Y., Albinet, A., Riffault, V., Chazeau, B., Amodeo, T., Salameh, D., Zhang, Y., Srivastava, D., Samaké, A., Aujay-Plouzeau, R., Papin, A., Bonnaire, N., Boullanger, C., Chatain, M., Chevrier, F., Detournay, A., Dominik-Sègue, M., Falhun, R., Garbin, C., Ghersi, V., Grignion, G., Levigoureux, G., Pontet, S., Rangognio, J.,

Zhang, S., Besombes, J.-L., Conil, S., Uzu, G., Savarino, J., Marchand, N., Gros, V., Marchand, C., Jaffrezo, J.-L., and Leoz-Garziandia, E.: Overview of the French Operational Network for In Situ Observation of PM Chemical Composition and Sources in Urban Environments (CARA Program), 12, 207, https://doi.org/10.3390/atmos12020207, 2021.

Petit, J.-E., Pallarès, C., Favez, O., Alleman, L. Y., Bonnaire, N., and Rivière, E.: Sources and Geographical Origins of PM10 in Metz (France) Using Oxalate as a Marker of Secondary Organic Aerosols by Positive Matrix Factorization Analysis, 10, 370, https://doi.org/10.3390/atmos10070370, 2019.

---

## Author Comment (AC2)

**Answer to referee #2**

In this manuscript, the authors presented source apportionment results of OP-DTT and OP-AA measured on $PM_{10}$ filter samples collected at 14 different locations in France using PMF and multiple linear regression models. The authors mainly focus on discussing the intrinsic OP, the variability of different sources, and the daily mean and median contribution of sources to OP. The limitation of the study is well discussed. This study has a unique dataset. However, in my opinion, the authors did not fully utilize their dataset. One of the uniqueness of this study is that the dataset spans 15 years and covers a wide range of environments. Seasonal variations of OP-DTT and OP-AA are discussed but the authors may consider looking into other aspects that are more interesting, for example, the spatial homogeneity or the historical changes of OP. How do the OP sources and source contributions change over the 15-year period? Are there differences in the historical trends of OP-DTT and OP-AA and how does it compare to PM mass? What chemical components are the most important drivers to the changes of OP and PM mass? It is well known that biomass burning, traffic emissions, secondary processing are important OP sources. Presenting something other than sources would enhance the scientific significance of this manuscript. Below are my comments:

We thank the reviewer for its positive feedback and numerous propositions!

Most of them are highly interesting and were considered in the revised version of the manuscript. However, some of them are impossible to achieve with the datasets used in the present study.

The spatial homogeneity of OP was already studied in a previous paper (Calas et al. 2019) using 7 sites in France. We agree that this temporal and spatial variability of OP could be further emphasized, but we chose in this article to focus on the source-apportionment of OP, that already raises lots of questions and discussions.

We do not think, however, that the current dataset could be used to investigate the temporal trend of OP over the region of interest. Indeed, many parameters are changing together (site location, sources identification and contribution, meteorology, etc). We here want to stress that the dataset does not cover a 15-year period, but 14 sampling sites with 1 to 2 years of sample collection. To clarify, we added figure in the supplementary information to better highlight the sampling period per site.

Major comments:

1. Please provide detailed protocols for both the DTT and AA assays. Multiple versions of DTT assays are currently used in the community. It is important to show which DTT protocol was used in this work. Perhaps the most important is the initial concentration of DTT as studies have shown that the initial DTT concentrations can have a large impact on the DTT consumption rates e.g. (Lin and Yu 2019).

The OP assays conducted at IGE have been deeply investigated and published as methodological papers (Calas et al, 2017, 2018).

For OP DTT, we use an initial concentration of 12.5 nmol of DTT (50 µL of 0.25 mM DTT solution in phosphate buffer) to react with 205 µL of phosphate buffer and 40 µL of PM suspension.

For OPAA, we mix 80 µL of PM suspension with 24 nmol of AA (100 µL of 0.24 mM AA solution in Milli-Q water) and follow the AA depletion within 30 min.

This has been added in the revised manuscript.

2. Line 110, particles removed from the filters were added to 96-well plates for DTT analyses, and the authors claimed that this included "soluble and insoluble" particles. What is the extraction efficiency? Could there be particles that are not extractable and attached to the filters. Other studies that measured total DTT run the extract along with the filter in DTT solutions. E.g. (Gao, Fang et al. 2017). A note should be added to emphasize the differences in the protocol and state that the DTT activities may not be "total".

We thank the referee for this remark. We already investigated this issue in the methodological paper of Calas et al. 2017: "The importance of simulated lung fluid (SLF) extractions for a more relevant evaluation of the oxidative potential of particulate matter". The efficiency of the extraction is quantified according to different protocols for extraction and according to the particle extraction media. Of course, it may vary according to the nature of particles. Then the supernatant (to avoid filter in the well) without filtration is injected in the wells leading to soluble and insoluble (visible) particles being in contact with DTT. We don't think this methodology can strongly differ from (Gao, Fang et al, 2017), and this previous extraction of particles from the filter is justifed by the use of the microplate reader that prevent to put the filter directly into the well (interfering with light source of the reader that would prevent from the online monitoring of the reaction).

[Figure]

[Figure]

[Figure]

**Figure S 11:** Kinetics of DTT depletion (normalized per µg of PM) when comparing three technics of extractions: vortex, sonication and a combined extraction (sonication followed by vortex). (Bars correspond to standard deviation (SD) of triplicate).

**Figure S 12:** Kinetics of DTT depletion when comparing the same sample extracted in Milli-Q water or Gamble + DPPC solution. (Bars correspond to SD of triplicate).

3. In Figure 3, OP-AA from road traffic, biomass burning, dust and OP-DTT from biomass burning and dust sources are bi-model distributions. Why? It would be interesting to look into details in chemical components to figure out the observed distributions.

Figure 3 presents the distribution of the 15 independent MLR model runs (one model analysis per site). Each series of model runs has been associated with uncertainty estimation using

bootstrap, leading to a Gaussian distribution of the regression coefficient (i.e. intrinsic OP of the sources). The multi-modal distribution observed in the Figure 3 reflect the individual Gaussian distribution obtained for each site.

Since this section has been re-written according to comments raised by reviewer#1 and next comment, some discussion is now provided for this variability. However, former Figure 3 has also been replaced by a more concise and general view of the intrinsic OP. We want here to emphasize the generalities of intrinsic OP, thanks to the extensive dataset, and not to discuss the details of each site (that could be done one by one potentially in a more specific study of each site). Former Figure 3 is now in SI alongside the table of intrinsic OP per site.

Some hypothesis concerning the multimodal distribution of intrinsic OP is now better described in the revised manuscript as well (difference in chemical compound or aging). See revised manuscript for the details since the multimodal reason doesn't seem to be unique but factor and site dependant.

4. Section 3.3.6-3.3.11, I appreciate the authors' efforts in discussing the variability of the intrinsic OP. However, it is not clear to me what do variabilities of different OP sources really bring about. It seems more interesting to compare the intrinsic OP or the contribution of different sources to OP with those from other studies or those from other regions of the world. These subsections lack in-depth discussion on each source. For example, for road traffic, transition metals (non-exhaust) and quinones from PAHs or soot (exhaust) can contribute to OP-DTT and OP-AA. How does your source profile from road traffic differ from others? What are the linkages of traffic-related chemical components to measured OP? What new insights does this work bring? One interesting questions is whether you can differentiate the contribution of exhaust and non-exhaust emissions to OP.

It would be, indeed, of great interest to compare this source-apportionment of OP to other studies. However, due to the differences in methodology to measure the OP or to estimate source contribution to PM mass, it is still difficult to directly compare intrinsic OP between studies using different protocols (use of gamble+DPPC, iso-mass, PMF-filter, CMB, PMF-AMS, PCA, etc). This study, however, presents a method of inter-comparison for homogeneous datasets and a national synthesis that could be useful for inter-study comparisons (i.e., other regions in the world) in the future.

Regarding the road traffic factor(s), we actually do not have a single factor here, but 15 different ones, determined by 15 different PMF (one for each studied site). They do present some variabilities in terms of chemical compound and are now discussed in the text and in the SI (with the similarity assessment) and in Weber et. al (2019). Comparing these factors with previous studies would be indeed of great interest. However, this goes far beyond the scope of this study (non-homogeneous dataset comparison, methodology of source apportionment, etc). For your interest, the chemical profiles used in this study aims to be uploaded to the SPECIEUROPE database, notably for comparison exercise. We were not able to differentiate the exhaust and non-exhaust of traffic-related emission in our current dataset. Hence, we cannot discuss their respective impact on OP.

We purposely did not discuss in depth the link between the chemical component of factor profile and the OP, because the whole point of doing the apportionment of OP by sources and not by species is to get rid of the issue of the all array of un-measured chemical species. For instance, doing inversion by species would lead to a high impact of levoglucosan, although this species has no effect on OP, simply due to the co-emission of other organic species with levoglucoan, or even due to formation of SOA (quinone-containing for instance) due to the aging of biomass burning emissions (see the recent paper of Campbell et al. 2021 for instance). « Hiding » everything into source categories allow to have only few molecular proxies of sources but still being able to identify important source contributors, even without knowing the exact source composition and responsible species.

5. It would be useful to present how are metals (especially Fe and Cu) apportioned into each factor. Atmospheric metals can be found in biomass burning, road traffic, dust, or sulfate particles by acid processing. Discussions on how metals are distributed in these factors may help to interpret the contribution of different sources to OP.

The full description of the source profiles is given in the interactive SI (http://getopstandop.u-ga.fr/results?component=pmf_profiles). We also extensively revised section 3.3 to discuss more in-depth the different profiles, with regards to both their chemical compound and variability across sites.

6. It is not clear that what are the new findings in this manuscript compared to previous work from the same group, e.g. (Weber, Uzu et al. 2018).

Weber et al, 2018, implemented the first OP apportionment with PMF tool in Chamonix as a proof of concept. In the present paper, we now expand this unified methodology to 15 sites datasets. The main point here is to investigate if what was previously established held at a larger geographical scale or rather if each site would present very different ranking of intrinsic OP per source. Since each site present similar ranking and order of magnitude for the intrinsic OP, what we presented earlier is not a singular case but present a "global approach" that synthesizes the geochemistry and OP of $PM_{10}$ in France.

In the present study, we also proposed different statistical tools used to quantify population exposure (the mean vs. median discussion). We highlighted that a drastic difference can be observed in this regard. With previously « limited » datasets, such an investigation could not be conducted due to low number of samples and poor statistical analysis possibility. Documenting the mean and median in this work is then an important step towards closing the gap between atmospheric science and epidemiological.

Other comments:

1. Line 257, "organic specie" should be "organic species"
2. Line 258, define "HULIS"
3. Use the same font type throughout, for example, line 464, numbers seem to be a different font than other contents

We thank the reviewer for their careful reading. For the font type, it is unfortunately due to the latex template of ACPD of unit formatting.

References:

Gao, D., T. Fang, V. Verma, L. Zeng and R. Weber (2017). "A method for measuring total aerosol oxidative potential (OP) with the dithiothreitol (DTT) assay and comparisons between an urban and roadside site of water-soluble and total OP." Atmos. Meas. Tech. Discuss. **2017**: 1-25.

Lin, M. and J. Z. Yu (2019). "Dithiothreitol (DTT) concentration effect and its implications on the applicability of DTT assay to evaluate the oxidative potential of atmospheric aerosol samples." Environmental Pollution **251**: 938-944.

Weber, S., G. Uzu, A. Calas, F. Chevrier, J. L. Besombes, A. Charron, D. Salameh, I. JeÅ¾ek, G. MoÄnik and J. L. Jaffrezo (2018). "An apportionment method for the oxidative potential of atmospheric particulate matter sources: application to a one-year study in Chamonix, France." Atmos. Chem. Phys. **18**(13): 9617-9629.

**References:**

Calas, A.; Uzu, G.; Besombes, J.-L.; Martins, J.M.F.; Redaelli, M.;  Weber, S.; Charron, A.; Albinet, A.; Chevrier, F.; Brulfert, G.; Mesbah,  B.; Favez, O.; Jaffrezo, J.-L. Seasonal Variations and Chemical  Predictors of Oxidative Potential (OP) of Particulate Matter (PM), for Seven Urban French Sites. Atmosphere 2019, 10, 698. https://doi.org/10.3390/atmos10110698

Calas, A., Uzu, G., Martins, J. M. F., Voisin, D., Spadini, L., Lacroix, T., and Jaffrezo, J.-L.: The importance of simulated lung fluid (SLF) extractions for a more relevant evaluation of the oxidative potential of particulate matter, Sci Rep, 7, 11617, https://doi.org/10.1038/s41598-017-11979-3, 2017.

Campbell, S. J., Wolfer, K., Utinger, B., Westwood, J., Zhang, Z.-H., Bukowiecki, N., Steimer, S. S., Vu, T. V., Xu, J., Straw, N., Thomson, S., Elzein, A., Sun, Y., Liu, D., Li, L., Fu, P., Lewis, A. C., Harrison, R. M., Bloss, W. J., Loh, M., Miller, M. R., Shi, Z., and Kalberer, M.: Atmospheric conditions and composition that influence PM2.5 oxidative potential in Beijing, China, 21, 5549–5573, https://doi.org/10.5194/acp-21-5549-2021, 2021.